# Dynamic Pricing and Learning
# with Bayesian Persuasion

**Shipra Agrawal**
Columbia University
sa3305@columbia.edu

**Yiding Feng**
University of Chicago
yidingfeng@uchicago.edu

**Wei Tang**
Columbia University
wt2359@columbia.edu

## Abstract

We consider a novel dynamic pricing and learning setting where in addition to setting prices of products in sequential rounds, the seller also ex-ante commits to 'advertising schemes'. That is, in the beginning of each round the seller can decide what kind of signal they will provide to the buyer about the product's quality upon realization. Using the popular Bayesian persuasion framework to model the effect of these signals on the buyers' valuation and purchase responses, we formulate the problem of finding an optimal design of the advertising scheme along with a pricing scheme that maximizes the seller's expected revenue. Without any apriori knowledge of the buyers' demand function, our goal is to design an online algorithm that can use past purchase responses to adaptively learn the optimal pricing and advertising strategy. We study the regret of the algorithm when compared to the optimal clairvoyant price and advertising scheme.

Our main result is a computationally efficient online algorithm that achieves an $O(T^{2/3}(m \log T)^{1/3})$ regret bound when the valuation function is linear in the product quality. Here $m$ is the cardinality of the discrete product quality domain and $T$ is the time horizon. This result requires some natural monotonicity and Lipschitz assumptions on the valuation function, but no Lipschitz or smoothness assumption on the buyers' demand function. For constant $m$, our result matches the regret lower bound for dynamic pricing within logarithmic factors, which is a special case of our problem. We also obtain several improved results for the widely considered special case of additive valuations, including an $\tilde{O}\left(T^{2/3}\right)$ regret bound independent of $m$ when $m \leq T^{1/3}$. [1]

## 1 Introduction

Dynamic pricing is a key strategy in revenue management that allows sellers to anticipate and influence demand in order to maximize revenue and/or utility. When the customer valuation and demand response for a product is apriori unknown, price variation can also be used to observe and learn the demand function in order to adaptively optimize price and revenue over time. This learning and optimization problem has been a focus of much recent literature that uses exploration-exploitation and multi-armed bandit techniques with dynamic pricing algorithms (e.g., see [43, 16, 41, 8]).

In practice, there is another important tool available to sellers in the form of *advertising*, using which the sellers can inform and shape customers' valuations of a product. It has been theoretically [53, 54] and empirically [55, 49] shown that advertisements can serve as a credible signal of the quality or characteristics of the advertised product. Sellers can use advertising to provide partial information about a product in order to better position the product in the market and potentially increase customers' chances of purchasing the product. For example, as a common strategy to drive

---

[1]A full version of this work is at `https://arxiv.org/abs/2304.14385`.

37th Conference on Neural Information Processing Systems (NeurIPS 2023).

subscriptions, online newspapers may use a "teaser" that selectively includes previews of some news articles that are likely to entice readers to subscribe for access to the full story; in the online used-car market, the dealer can advertise the used car by emphasizing different aspects of the car, such as fuel efficiency/mileage/unique features, or selectively disclose history-report information from reputable third parties, catering to specific buyer interests; a film distributor may advertise the movie by selectively showing footage from the film.

However, advertising must be carefully designed to achieve the desired gains. At one glance up-selling or inflating the product quality by selectively disclosing only favorable information might appear as a profitable advertising strategy. But such strategies carry the disadvantage of not being very effective in modifying customer beliefs as customers may not trust that the provided information accurately reflects the product's true quality. Also, the design of the advertising strategy needs to interact with the design of the pricing strategy and account for the demand function. For example, to sell highly-priced products or under heavy competition/low demand, the customer may need to be convinced of a good match through more information and thorough insights about the product characteristics. On the other hand, in markets with high demand or for very low-priced goods, the seller may get away with revealing very little information. An extreme example of this phenomenon is the concept of *mystery deal boxes* sold by some retailers like Amazon/Woot, where the customers are not even made aware of the exact contents of the low-cost box that they are purchasing.[2]

In this paper, we use a *Bayesian persuasion framework* [39] to model the effect of an advertising strategy on customers' beliefs about product quality and consequently their purchase decisions. Our novel formulation combines the Bayesian persuasion model with dynamic pricing and learning in order to quantify the tradeoffs between the design of the pricing and advertising strategies and their combined impact on the revenue outcomes. Without any apriori knowledge of the demand function, our goal is to design an online algorithm that can use past customer responses to adaptively learn a joint pricing and advertising strategy that maximizes the seller's revenue.

Bayesian persuasion is a popular framework for information design with several different settings considered in the literature [39, 30, 38, 10]. We consider a Bayesian persuasion model where the sender (seller) ex-ante commits to an information policy (advertising strategy) that prescribes the distribution of signals the sender will provide to the receiver (buyer) on observing the true state of the world (product quality). The receiver, on observing the sender's signal, uses Bayes' rule to form a posterior on the state of the world. The receiver's action (purchase decision) then optimizes their expected utility under this posterior.

**Problem formulation.** Specifically, our *dynamic pricing and advertising problem* is formulated as follows. There are $T$ sequential and discrete rounds. In each round, a fresh product is offered by the seller, with a public prior distribution $\lambda$ on the product quality $\omega \in \Omega \subseteq [0, 1]$. At the beginning of each round $t$, before observing the realized quality of the $t^{th}$ product, the seller commits to a price $p_t \in [0, U]$ and an advertising strategy $\phi_t$, where $\phi_t(\sigma|\omega)$ prescribes the distribution of signal $\sigma \in \Sigma$ where $\Sigma$ is an arbitrary signaling space given the realized product quality $\omega \in \Omega$.

A buyer arrives in each round $t$ with *private* type $\theta_t$ generated i.i.d. from a distribution with CDF $F(\cdot)$ and support[3] $\Theta = [0, 1]$. The CDF $F(\cdot)$ (or equivalently the demand function $D(\cdot) \triangleq 1 - F(\cdot)$) is fixed but unknown to the seller. For a buyer of type $\theta \in \Theta$, the valuation of a product with product quality $\omega$ is given by function $v(\theta, \omega)$.

The $t^{th}$ product quality $\omega_t \sim \lambda$ is then realized and observed by the seller. The buyer cannot observe the realized product quality, but only a signal $\sigma_t \sim \phi_t(\cdot|\omega_t)$ provided by the seller. The buyer uses this signal along with the prior $\lambda$ to formulate a Bayesian posterior distribution on the product quality $\mu_t(\omega|\sigma_t) \propto \phi_t(\sigma_t|\omega) \cdot \lambda(\omega)$. The buyer then purchases the product if and only if the expected valuation under this posterior $\mathbb{E}_{\omega \sim \mu_t(\cdot|\sigma_t)}[v(\theta_t, \omega)]$ is greater than or equal to the price $p_t$. We denote the buyer decision at time $t$ by $a_t \in \{0, 1\}$ with $a_t = 1$ denoting purchase.

**Remark 1.1.** *We consider the setting where buyers use only the prior distribution and signal in the current round, and not the signals in the past rounds, to make their decisions. This is motivated by*

---

[2]For example, when selling the opaque products, the precise product features or characteristics are hidden from the customers [31].

[3]Our results can be generalized to the setting where $\Theta$ is any compact interval $[\underline{\theta}, \bar{\theta}] \subseteq \mathbb{R}^+$, or unbounded (see Remark 2.3).

*the fact that at each round, the buyer is facing a fresh product, whose quality is drawn independently across time. Thus, previous signals do not carry information about the current product. Fresh products with independent qualities are common in many real-world applications such as second-hand markets, mystery boxes sold by Amazon/Woot, etc. This similar problem structure has also been studied in the online/sequential Bayesian persuasion literature with repeated interactions of sender and receivers, for example, see [61, 24, 23, 33, 58, 15].*

A summary of the game timeline is as follows: at $t \in [T]$, (1) the seller commits to a price $p_t \in [0, U]$ and an advertising strategy $\phi_t$; (2) a buyer $t$ with private type $\theta_t \sim F$ arrives; (3) a product with quality $\omega_t \sim \lambda$ is realized; the seller sends signal $\sigma_t \sim \phi_t(\cdot|\omega_t)$ to the buyer; (4) the buyer formulates Bayesian posterior $\mu_t(\omega|\sigma_t)$, and the buyer purchases the product (denoted by $a_t = 1$) to generate revenue $p_t$ if and only if her expected value exceeds the price, i.e., $\mathbb{E}_{\omega \sim \mu_t(\cdot|\sigma_t)}[v(\theta_t, \omega)] \geq p_t$.

Our goal is to design an online learning algorithm that sequentially chooses the price and advertising strategy $p_t, \phi_t$ in each round $t$ based on the buyers' responses in the previous rounds, in order to optimize total expected revenue over a time horizon $T$ without apriori knowledge of the distribution $F$. Let $\mathrm{Rev}(p_t, \phi_t) \triangleq \mathbb{E}[p_t \cdot a_t; p_t, \phi_t]$ denote the expected revenue at time $t$ under the price $p_t$ and advertising strategy $\phi_t$. Here the expectation is over realizations of customer type $\theta_t \sim F$, product quality $\omega_t \sim \lambda$ and advertising signal $\sigma_t \sim \phi_t(\cdot|\omega_t)$. Note that since product quality and types are i.i.d. across time, for any given $p_t = p, \phi_t = \phi$, the expected per-round revenue $\mathrm{Rev}(p, \phi)$ does not depend on time. Thus a static price and advertising strategy maximizes total expected revenue over the time horizon $T$. Therefore, we can measure the performance of an algorithm in terms of *regret* that compares the total expected revenue of the algorithm to that of the best static pricing and advertising strategy. We define

$$\mathrm{Regret}(T) \triangleq T \max_{p, \phi} \mathrm{Rev}(p, \phi) - \sum_{t=1}^{T} \mathrm{Rev}(p_t, \phi_t) .$$

**Our contributions.** In this work, we present a computationally efficient online pricing and advertising algorithm that achieves an $O\left(T^{2/3}(m \log T)^{1/3}\right)$ bound on the regret in time $T$, where $m = |\Omega|$ is the cardinality of the (discrete) product quality space. Importantly, we achieve this result without any assumptions like Lipschitz or smoothness on the demand function $D(\cdot) = 1 - F(\cdot)$. However, our results require certain assumptions on the valuation function. Following the literature, we make the common assumption that the function $v(\theta, \omega)$ is linear in the product quality $\omega$. Furthermore, we assume the following monotonicity and Lipschitz properties on the valuation function.

**Assumption 1.** *Buyer's valuation function $v(\cdot, \cdot)$ satisfies:*

**1a** *Fix any buyer type $\theta$, function $v(\theta, \omega)$ is non-decreasing w.r.t. quality $\omega$.*

**1b** *Fix any quality $\omega$, function $v(\theta, \omega)$ is increasing and 1-Lipschitz[4] w.r.t. type $\theta$.*

Such assumptions are in fact common in literature and natural in many economic situations where the valuation of a product increases with the product quality and buyer's type (paying ability/need). Existing literature on Bayesian persuasion/dynamic pricing often makes even stronger assumptions about the receiver's utility function. For example, both the additive functions $v(\theta, \omega) = \theta + \omega$ [cf. 35, 45, 27] and the multiplicative functions $v(\theta, \omega) = \theta\omega + \theta$ [cf. 22, 48], are linear in $\omega$ and satisfy Assumption 1. Our main result is then summarized as follows:

**Theorem 1.1.** *For any type CDF $F$, given a valuation function $v(\theta, \omega)$ that is linear in product quality $\omega$ and satisfies Assumption 1, Algorithm 1 with parameter $\varepsilon = \Theta((^{m \log T}/_T)^{1/3})$ has an expected regret of $O\left(T^{2/3}(m \log T)^{1/3}\right)$. Here, $m$ is the cardinality of the discrete quality space $\Omega$.*

Furthermore, we obtain several improved results for the widely considered special case of additive valuations, i.e., for $v(\theta, \omega) = \theta + \omega$. See Appendix B for the formal statements and analysis.

1. **(Theorem B.1)** Consider discrete sets $\Omega$ that are 'equally-spaced', e.g., $\Omega = \{0, 1\}$ or $\Omega = [m]$. Given such a product quality space $\Omega$ and additive valuation function, we show that the regret of Algorithm 1 is bounded by $O(T^{2/3}(\log T)^{1/3})$ when $m \leq (T/\log T)^{1/3}$, and by $O(\sqrt{mT \log T})$ for larger $m$.

---

[4]Here 1 Lipschitz constant is for exposition simplicity, an arbitrary Lipschitz constant $L$ can be treated similarly.

2. **(Theorem B.2)** For any arbitrary (discrete or continuous) product quality space $\Omega$, given additive valuation functions, we have a slightly modified algorithm (see Algorithm 3 in Appendix C.2) with an expected regret of $O(T^{3/4}(\log T)^{1/4})$ independent of $m$.

When the valuation function $v(\cdot, \omega)$ is $L$-Lipschitz w.r.t. type $\theta$, our regret bound in Theorem 1.1 would become $O(T^{2/3}(Lm \log T)^{1/3})$, and the regret bound for arbitrary space $\Omega$ would become $O(T^{3/4}(L \log T)^{1/4})$.

We might compare our results to the best regret bounds available for the well-studied dynamic pricing and learning problem with unlimited supply [43, 8], which is a special case of our problem if the product quality is deterministic, i.e., $m = 1$, and the advertising scheme reveals no information and thus has no impact on the buyer's purchase decision. For the dynamic pricing problem a lower bound of $\Omega(T^{2/3})$ on regret is known [43]. Therefore the dependence on $T$ in our results cannot be improved. In fact, our result matches this lower bound in the case of binary or constant size quality space, which are common settings in information design literature [39, 18, 5, 57, 33, 12].

**High-level descriptions of the proposed algorithm and challenges.** Our problem can be viewed as a very high dimensional combinatorial multi-armed bandit problem, where each arm is a pair of a price and a feasible advertising strategy: the set of feasible advertising strategies being the set of all possible conditional distributions $\{\phi(\cdot|\omega), \omega \in \Omega\}$ over signal space $\Sigma$. As a first step towards obtaining a more tractable setting, we present an equivalent reformulation of the problem which uses the observation that advertising affects the buyer's decision only via the posterior distribution over quality. By the linearity of valuation function $v(\cdot, \cdot)$ over product quality $\omega$, seller's choice of advertising strategy in every round can be further simplified to selecting a distribution over posterior means that is subject to a feasibility constraint.

From here, the seller's decision space now becomes two-dimensional (a price and a distribution of posterior means). Viewing seller's expected revenue as an unknown (nonlinear) function over this two-dimensional decision space, one may consider applying algorithms in contextual bandits or Lipschitz bandits to get sublinear regrets, e.g., $\tilde{O}(T^{3/4}\mathsf{poly}(m))$ regret for two-dimensional decision space if there exists a Lipschitz property of reward function relative to seller's decision space. However, it is unclear whether one can establish such Lipschitz property given that we do not assume Lipschitzness or smoothness on demand function and we have complex constraints on the feasibility of advertising space. Instead, in our algorithm we use a 'model-based approach': we use buyers' purchase responses to explore the demand function over the (discretized) type space and jointly learn the optimal advertising and pricing. To explore the demand function over the one-dimensional (discretized) type space, we propose a novel discretization scheme such that it enables near-optimal price and advertising strategy even without Lipschitzness or smoothness assumption and with the complex feasibility constraint on the advertising space. These treatments lead us to the optimal $\tilde{O}(T^{2/3}m^{1/3})$ regret.

**Related work.** Our work is related to several streams of research. Below we briefly review the some of these connections.

In our setting, the seller can utilize her information advantage to design an advertisement to signal the product quality to the buyer. There is a long line of research in the literature, from both empirical and theoretical perspective, dedicated to study how to use advertisement as a signal to steer buyers' evaluations of advertised goods [53, 54, 42, 52, 37, 55, 40]. In our problem, we follow the literature in information design, a.k.a., Bayesian persuasion [39] [also see the recent surveys by 30, 38, 10], where the seller can commit to an information policy that can strategically disclose product information to the buyer so that to influence buyer's belief about the product quality. Similar formulation for advertising has also appeared in [19, 32, 3, 34, 14, 13, 28]. Our work differs from these works in several ways. First, the seller's offline problem in our setting is a joint pricing and advertising problem. Second, we focus on an online setting where the seller has no apriori knowledge of the demand function and has to use past buyers' purchase responses to adaptively learn optimal pricing and advertising strategy.

Our problem shares similarity to the problem on sale of information in economics and computer science literature [7, 9, 11, 26, 20, 48, 12, 59, 47, 25]. In particular, similar to the problem on sale of information, the seller, in our setting, also commits to design an information structure to reveal information about the realized state to the decision maker (i.e., buyer); and the buyer, in our setting,

then makes the payment based on the declared information structure, not for specific realizations of the seller's informative signals [12, 20, 48]. However, different from these works, the seller in our setting is selling a product with some inherent value and not just information. The valuation of the product can be shaped by providing information. This gives new interesting tradeoffs between information and revenue in our problem that are absent in the settings where only information is being sold. Moreover, in most literature of selling information, the buyers' type distribution is usually assumed to be known. We consider a more practical data-driven setting where the underlying buyers' type distribution (demand function) is apriori unknown to the seller and needs to be learnt from observations.

Facing unknown buyer's preference (i.e., buyer's private type), the seller's dynamic advertising problem also relates to the growing line of work in information design on relaxing one fundamental assumption in the canonical Bayesian persuasion model – the sender perfectly knows receiver's preference. The present paper joins the recent increased interests on using online learning approach to study the regret minimization when the sender repeatedly interacts with receivers [23, 24, 61, 33] without knowing receivers' preferences. Moreover, our work also conceptually relates to research on Bayesian exploration in multi-armed bandit [46, 51, 50, 36] which also studies an online setting where one player can utilize her information advantage to persuade another player to take the desired action. Our work departs from this line of work in terms of both the setting and the application domain. Particularly, the above works typically consider an online setting on how to learn optimal signaling scheme whereas in our setting the optimal policy is a joint pricing and signaling (advertising) scheme.

When there is no uncertainty in the product quality, the seller's problem in our setting reduces to a standard dynamic pricing and learning problem with unknown non-parametric demand function [43, 16, 41, 8]. However, given any non-trivial product quality space and prior distributions, in our problem, in addition to a price, the seller needs to choose a non-trivial advertising strategy in order to maximize revenue. This makes the seller's decision space high dimensional and (as we discuss in the next section) introduces significant complexities and difficulties so that the typical techniques (like uniform or adaptive discretization) used in pricing and continuous/combinatorial multi-armed bandit literature cannot be directly applied.

## 2 Algorithm Design

In this section, we present our main algorithm for the dynamic pricing and advertising problem. In subsection 2.1, we present an equivalent reformulation for tractable advertising strategies, then in subsection 2.2, we discuss many important challenges even after this simplification, and finally, in subsection 2.3 we present our algorithm.

### 2.1 An equivalent reformulation for tractable advertising strategies

Recall that in every round, the buyer $t$ sees the offered price $p_t$ and advertising strategy $\phi_t$ that specifies the distributions over signals $\phi_t(\cdot|\omega) \in \Delta^\Sigma$ that the seller will send for each possible value $\omega$ of the realized product quality. After the product quality $\omega_t$ is realized, the buyer $t$ sees a signal $\sigma_t \sim \phi_t(\cdot|\omega_t)$ from the seller's declared advertising strategy. The buyer uses this signal along with the prior $\lambda$ to form a Bayesian posterior $\mu_t(\cdot|\sigma_t) \in \Delta^\Omega$ on the product quality. The Bayesian rational buyer then takes the action $a_t \in \{0, 1\}$, based on expected utility maximization. In particular, we have $a_t = 1$ if $\mathbb{E}_{\omega \sim \mu_t(\cdot|\sigma_t)}[v(\theta_t, \omega)] \geq p_t$ and $a_t = 0$ otherwise.

From the decision formula above, it is clear that the choice of advertising strategy affects the buyer $t$'s decision only through the realized posterior $\mu_t(\cdot|\sigma_t)$. Consequently, the seller's choice of advertising strategy in time $t$ can be reduced to selecting a distribution over posteriors $\mu_t$. Seller's choice can in fact be further simplified in the case where the buyer's valuation function is linear in the product quality $\omega$ since in that case we have

$$\mathbb{E}_{\omega \sim \mu_t(\cdot|\sigma_t)}[v(\theta_t, \omega)] = v(\theta_t, \mathbb{E}_{\omega \sim \mu_t(\cdot|\sigma_t)}[\omega]) = v(\theta_t, q_t)$$

where $q_t$ is the realized posterior mean $q_t \triangleq \mathbb{E}_{\omega \sim \mu_t(\cdot|\sigma_t)}[\omega]$. Here $q_t \in [0, 1]$ since $\omega \in \Omega \subseteq [0, 1]$. Therefore the buyer purchases ($a_t = 1$) if and only if $v(\theta_t, q_t) \geq p_t$. As a result, we can reduce the seller's advertising in round $t$ to the choice of distribution (pdf) $\rho_t(\cdot) \in \Delta^{[0,1]}$ over posterior means.

However, the choice of $\rho_t$ must be restricted to only feasible distributions of posterior means, that is, all possible distributions over posterior means that can be induced by any advertising scheme given the prior $\lambda$. It is well known that the distribution over posterior means $\rho$ is feasible if and only if it is the mean-preserving contraction of the prior [17, 6]. This condition can be equivalently written in terms of a Bayes-consistency condition [39] on the conditional means $\rho(\cdot|\omega), \omega \in \Omega$. For simplicity of exposition, we consider discrete quality space $\Omega = \{\bar{\omega}_1, \ldots, \bar{\omega}_m\} \subseteq [0, 1]$, where $0 = \bar{\omega}_1 < \bar{\omega}_2 < \ldots < \bar{\omega}_m = 1$ and $m = |\Omega|$ is the cardinality of the quality space. Then, a distribution $\rho$ over posterior means is feasible if one can construct a set of conditional distributions $(\rho_i)_{i \in [m]}$ satisfying the following Bayes-consistency condition, and vice versa [39]:

$$\frac{\sum_{i \in [m]} \lambda_i \rho_i(q) \bar{\omega}_i}{\sum_{i \in [m]} \lambda_i \rho_i(q)} = q, \quad \forall q \in \mathsf{supp}(\rho) . \tag{BC}$$

Throughout this paper, we use the collection of distributions $(\rho_i)_{i \in [m]}$ satisfying (BC) condition as a convenient way to construct feasible distributions over posterior means: $\rho(q) = \sum_i \rho_i(q) \lambda_i$.

With the above observations, we can without loss of generality assume that seller's advertising strategy is to directly choose a distribution $\rho_t$ over the posterior means that satisfies (BC), without considering the design of the underlying signaling scheme $\{\phi(\sigma|\omega), \Sigma\}$.

We summarize the new equivalent game timeline as follows: at $t \in [T]$, (1) the seller commits to a price $p_t \in [0, U]$ and an advertising strategy $\rho_t = (\rho_{i,t})_{i \in [m]}$ satisfying (BC); (2) a buyer $t$ with private type $\theta_t \sim F$ arrives; (3) a product with quality $\omega_t \sim \lambda$ is realized; a posterior mean $q_t \sim \rho_t$ is realized; (4) the buyer observes the posterior mean $q_t$; the buyer purchases the product ($a_t = 1$) to generate revenue $p_t$ if only if $v(\theta_t, q_t) \geq p_t$. Note that the seller knows the form of the buyer's valuation function $v$. Moreover, the seller observes the realized product quality $\omega_t$, the realized posterior mean $q_t$, and the buyer's purchase decision $a_t$, but does not know type CDF $F$ (i.e., the demand function $D$) and the realized buyer type $\theta_t$.

**Revenue and regret.** Given the new formulation, we can also rewrite the revenue and regret in terms of the choices of $\rho_t, t = 1, \ldots, T$. We define the following function $\kappa(p, q)$, which we refer to as the *critical type* for a given price $p$ and posterior mean $q$.

**Definition 2.1** (Critical type). *For any $p \in [0, U]$ and $q \in [0, 1]$, define function $\kappa(\cdot, \cdot)$ as $\kappa(p, q) \triangleq \min\{\theta \in \Theta : v(\theta, q) \geq p\}$.*

Now under Assumption 1b, due to the monotonicity of the valuation function in buyer's type, we have that given any $p, q, \theta, \mathbf{1}[v(\theta, q) \geq p] = \mathbf{1}[\theta \geq \kappa(p, q)]$. Therefore, the buyer $t$ will purchase the product if and only if $\theta_t \geq \kappa(p_t, q_t)$.

Therefore, given the price, advertising $p_t = p, \rho_t = \rho$ and prior distribution $\lambda$, the expected revenue in any round $t$ is given by [5]

$$\mathsf{Rev}(p, \rho) = \mathbb{E}_{\omega \sim \lambda, \theta \sim F, q \sim \rho}[p \cdot \mathbf{1}[\theta \geq \kappa(p, q)]] = p \sum_i \lambda_i \int_0^1 \rho_i(q) D(\kappa(p, q)) dq \tag{1}$$

Let the seller's online policy offer price $p_t$ and advertising $\rho_t$ at time $t$, where $p_t, \rho_t$ can depend on the history of observations/events up to time $t$. Then expected regret defined in Section 1 can be equivalently written as

$$\mathsf{Regret}[T] = T\mathsf{Rev}(p^*, \rho^*) - \sum_{t=1}^{T} \mathbb{E}[\mathsf{Rev}(p_t, \rho_t)] .$$

Here, the expectation is taken with respect to any randomness in the algorithm's choice of $p_t, \rho_t$; and $p^*, \rho^* = (\rho_i^*)_{i \in [m]}$ are defined as the best price and advertising strategy for a given $F$ (and $\kappa(\cdot, \cdot)$ which is determined by $F$). Given the expression for $\mathsf{Rev}(p, \rho)$ derived above, these can be characterized by the following optimization program

$$p^*, \rho^* \triangleq \arg\max_{p, \rho} \ \mathsf{Rev}(p, \rho)$$

$$\text{s.t.} \quad \frac{\sum_{i \in [m]} \lambda_i \rho_i(q) \bar{\omega}_i}{\sum_{i \in [m]} \lambda_i \rho_i(q)} = q, \quad q \in [0, 1]; \quad \rho_i \in \Delta^{[0,1]}, \quad i \in [m] . \tag{$P_{\mathsf{OPT}}$}$$

where the first constraint is due to (BC).

---

[5]Here, we slightly abuse the notation to redefine Rev as a function of price and $\rho$, instead of price and $\phi$ defined earlier.

## 2.2 Algorithm design: challenges and ideas

**Challenge: high-dimensional continuous decision space.** In the last section, we obtained a considerable simplification of the problem by reducing the seller's advertising strategy in every round $t$ to a *distribution* $\rho_t \in \Delta^{[0,1]}$ over posterior means satisfying the (BC) condition. However, the decision space (a.k.a space of arms) still remains high dimensional and therefore a naive application of (uniform or adaptive) discretization-based bandit techniques, e.g. from Lipschitz bandit literature [44, 56], would not achieve the desired results.

**Algorithm design idea: exploring over one-dimesnional type space.** Our algorithm uses a 'model-based approach' instead, where we use buyer purchase responses to develop (upper confidence bound) estimates of the demand model $D(\theta) = 1 - F(\theta)$ on the points of a *discretized type space* $\mathcal{S} \subseteq \Theta$. We then use these upper confidence bounds to construct a piecewise-constant demand function that is an upper confidence bound (UCB) for the demand function $D$. Then, in each round we solve for the optimal price and advertising strategy by solving an optimization problem similar to $P_{OPT}$, but with the UCB demand function.

**Challenge: efficient discretization of type space.** The challenge then is to design a discretization scheme for the type space such that we have a) efficient learning, i.e., the discretized space can be efficiently explored to accurately estimate the demand function on those points, and b) Lipschitz property, i.e., the optimal revenue with the UCB estimate of demand function is close to the true optimal revenue as long as the estimation error on the discretized space is small.

There are two main difficulties in achieving this: (1) Lack of any smoothness/Lipschitz assumption on CDF $F$. (2) Sensitivity of the Bayes-consistency condition (BC). To see these difficulties, recall that given a price $p_t$ and realized posterior mean $q_t$, the $t^{th}$ buyer's purchase decision is given by $a_t = \mathbf{1}[\theta_t \geq \kappa(p_t, q_t)]$; thus the seller can obtain demand function estimate at point $\kappa(p_t, q_t)$. Without any smoothness or Lipschitz property of demand function, estimates of demand function cannot be extrapolated accurately to other points. This means that in our revenue optimization problem (estimated version of $P_{OPT}$), we need to find a price and advertising strategy that we can only use estimates of demand function on a discretized type space, say $\mathcal{S} \subseteq \Theta$. However, if we restrict to a discretized type space $\mathcal{S}$, then the support of posterior mean distributions (a.k.a advertising strategy) must be restricted to the points $q$ such that the corresponding critical types $\kappa(p, q)$ are in the set $\mathcal{S}$.

Unfortunately, the (BC) condition makes the set of feasible advertising strategies very sensitive to their support. In particular, if we use uniform-grid based discretization (which is commonly used in previous dynamic pricing literature such as [43, 8]), it is easy to construct examples of prior distribution and valuation function such that there are no or very few feasible advertising strategies with the corresponding restricted support.

**Example 2.2.** *Consider additive valuation function, i.e., $v(\theta, \omega) = \theta + \omega$, and thus $\kappa(p, q) = p - q$. Consider quality space $\Omega = \{\bar{\omega}_i\}_{i \in [3]}$. Given a small $\varepsilon$, consider a uniform-grid based discretization for the type space $\mathcal{S}$, i.e., $\mathcal{S} = \{0, \varepsilon, 2\varepsilon, \ldots, 1\}$. If we also use a price $p$ that is from uniform-grid based discretized price space, i.e., $p = k\varepsilon$ for some $k \in N^+$, then to ensure $\kappa(p, q) \in \mathcal{S}$, the support of advertising strategy (i.e., the distribution of the posterior means) must also be restricted within the set $\mathcal{S}$. However, if the prior distribution $\lambda$ has negligible probabilities on qualities $\bar{\omega}_1, \bar{\omega}_3$, and quality $\bar{\omega}_2 \notin \mathcal{S}$, then we cannot construct any posterior distribution with the mean in the set $\mathcal{S}$. Therefore, there does not exist any feasible advertising strategy.*

*Note that this difficulty cannot be fixed by simply modifying the discretized type space to $\mathcal{S} \cup \Omega$, because even then we would need to construct an advertising such that $\kappa(p, q) = p - q \in \mathcal{S} \cup \Omega$ in the grid for all $p$. That would need that the support of the strategy (i.e., posterior means $q$) is restricted to be in $\{k\epsilon - \bar{\omega}_i\}_{k \in \mathbb{N}^+, i \in [3]}$; such posterior means again may not be achieved here.*

**Algorithm design idea: novel discretization scheme.** Our algorithm employs a carefully-designed *quality-and-price-dependent discretization* scheme. The above example shows that we cannot uniformly discretize price and type using $\varepsilon$-grids, as we may not have any feasible advertising strategy under such discretization. And furthermore, it also shows that this difficulty cannot be fixed by simply adding the $m$ points in quality space to the discretized type space $\mathcal{S}$. Instead, in our discretization scheme, we first uniformly discretize the price space to an $\varepsilon$-grid $\mathcal{P}$. Then to construct a discretized

type space $\mathcal{S}$, in addition to the points on an $\varepsilon$-grid over $[0, 1]$, we also include points $\{\kappa(p, \omega)\}$ for every price $p \in \mathcal{P}$ and quality $\omega \in \Omega$. This gives us a discretized type space $\mathcal{S}$ of size $mU/\varepsilon$. We claim that our construction ensures that there exist near-optimal price and advertising strategy with support in the discretized type space $\mathcal{S}$. The proof of this claim requires a careful rounding argument, which forms one of the main technical ingredients for our regret analysis in Section 3.

## 2.3 Details of the proposed algorithm

Our dynamic pricing and advertising algorithm jointly discretizes the price space and type space using the following *quality-and-price-dependent discretiztion* scheme: given parameter $\varepsilon$, we define the following set:

$$\begin{aligned}
\mathcal{P} &\triangleq \{\varepsilon, 2\varepsilon, \ldots, U\} \\
\mathcal{S} &\triangleq \{0, \varepsilon, \ldots, 1 - \varepsilon, 1\} \cup \{(\kappa(p, \omega) \wedge 1) \vee 0\}_{p \in \mathcal{P}, \omega \in \Omega} .
\end{aligned} \tag{2}$$

At time $t$, we restrict the price and advertising strategies $(p_t, \rho_t)$ to the set of $(p, \rho = (\rho_i)_{i \in [m]})$ such that $p \in \mathcal{P}$, and given price $p$, each conditional distribution $\rho_i$ has restricted support $\mathcal{Q}_p$ defined as $\mathcal{Q}_p \triangleq \{q : \kappa(p, q) \in \mathcal{S}\}$. Given the price and advertising $p_t, \rho_t$, let the realized posterior mean at time $t$ be $q_t \sim \rho_t$, and let the corresponding critical type be $x_t \triangleq \kappa(p_t, q_t) \in [0, 1]$. Then, note that the above restrictions on price and advertising strategies guarantee that $x_t \in \mathcal{S}$.

Next, to compute the offered price and advertising strategy in round $t$, we optimize an upper confidence bound on the revenue function that we develop using upper confidence bounds $D^{\mathsf{UCB}}(x), x \in \mathcal{S}$ of the demand function computed as follows. For every type $x \in \mathcal{S}$, let $\mathcal{N}_t(x)$ denote the set of time rounds before time $t$ that the induced critical type is exactly $x$, and let $N_t(x)$ be the number of such time rounds. That is, $\mathcal{N}_t(x) \triangleq \{\tau < t : \kappa(p_\tau, q_\tau) = x\}, N_t(x) \triangleq |\mathcal{N}_t(x)|, x \in \mathcal{S}$.

Recall that buyer's purchase decision follows $a_\tau = \mathbf{1}[\theta_\tau \geq \kappa(p_\tau, q_\tau)]$. We estimate the demand function at $x$ as: $\bar{D}_t(x) \triangleq \frac{\sum_{\tau \in \mathcal{N}_t(x)} a_\tau}{N_t(x)}$. We can now define the following UCB index:

$$D_t^{\mathsf{UCB}}(x) = \min_{x' \in \mathcal{S}: x' \leq x} \bar{D}_t(x') + \sqrt{\frac{16 \log T}{N_t(x')}} + \frac{\sqrt{(1 + N_t(x')) \ln(1 + N_t(x'))}}{N_t(x')} \wedge 1, \quad x \in \mathcal{S} \tag{3}$$

Then, for any pair of discretized price $p \in \mathcal{P}$ and advertising strategy with discretized support for that price $\rho = (\rho_i \in \Delta^{\mathcal{Q}_p}, i \in [m])$, we define the following seller's revenue estimates:

$$\mathsf{Rev}_t^{\mathsf{UCB}}(p, \rho) \triangleq p \sum_{i \in [m]} \lambda_i \int_0^1 \rho_i(q) D_t^{\mathsf{UCB}}(\kappa(p, q)) dq .$$

Above is well-defined since by definition $\kappa(p, q) \in \mathcal{S}$ for each such $(p, q) \in \mathcal{P} \times \mathcal{Q}_p$. Finally, we let $p_t, \rho_t$ be the optimal solution to the following optimization problem:

$$(p_t, \rho_t) = \arg \max_{p, \rho} \mathsf{Rev}_t^{\mathsf{UCB}}(p, \rho)$$

$$\text{s.t.} \quad p \in \mathcal{P}; \quad \frac{\sum_{i \in [m]} \lambda_i \rho_i(q) \bar{\omega}_i}{\sum_{i \in [m]} \lambda_i \rho_i(q)} = q, \; q \in \mathcal{Q}_p; \; \rho_i \in \Delta^{\mathcal{Q}_p}, \; i \in [m] . \tag{$\mathsf{P}_t^{\mathsf{UCB}}$}$$

We summarize our algorithm as Algorithm 1. The main computational bottleneck of Algorithm 1 is to solve the high-dimensional program $\mathsf{P}_t^{\mathsf{UCB}}$ at each time $t \geq |\mathcal{S}| + 1$. As we illustrate in Proposition 2.1, there exists an efficient method to optimally solve this program. The proof of this result utilizes the monotoncity of the function $D_t^{\mathsf{UCB}}$.

**Proposition 2.1** (Adopted from [4, 21]). *Let $\varepsilon$ be the discretization parameter for the set $\mathcal{P}$ defined in (2). There exists a polynomial time (in $|\mathcal{S}|U/\varepsilon$) algorithm that can solve the program $\mathsf{P}_t^{\mathsf{UCB}}$.*

*Proof.* Since function $D_t^{\mathsf{UCB}}$ is monotone with discontinuities at the points in the set $\mathcal{S}$, when we fix a price $p \in \mathcal{P}$, the function $D_t^{\mathsf{UCB}}(\kappa(p, \cdot))$ is also monotone with discontinuities at the points in $\mathcal{Q}_p = \{q : \kappa(p, q) = x\}_{x \in \mathcal{S}}$. Given a prior $\lambda$, optimizing a monotone function with discontinuities over all feasible advertising strategies induced from the prior $\lambda$ subject to the constraint where the support of advertising strategies must be in the set $\mathcal{Q}_p$ has been studied in [4, 21]. It has been shown that there

---

**Algorithm 1:** Algorithm for Dynamic Pricing and Advertising with Demand Learning.

---

**1** **Input:** Discretization parameter $\varepsilon$.

**2** For the first $|\mathcal{S}|$ rounds, for each $x \in \mathcal{S}$, offer a price $p$ with no information advertising s.t.
$\kappa(p, \mathbb{E}_{\omega \sim \lambda}[\omega]) = x$. `// No information advertising provides completely`
`uninformative signal - the distribution` $\phi(\cdot|\omega)$ `of signals does not depend on the`
`realized quality` $\omega$

**3** **for** *each round* $t = |\mathcal{S}| + 1, |\mathcal{S}| + 2, \ldots, T$ **do**

**4** $\quad$ For all $x \in \mathcal{S}$, compute $D_t^{\mathsf{UCB}}(x)$ as defined in (3).

**5** $\quad$ Offer the price $p_t$ and an advertising $\rho_t$ computed as an optimal solution to program $\mathsf{P}_t^{\mathsf{UCB}}$.
$\quad$ `/*` $p_t, \rho_t$ `satisfies` $\kappa(p_t, q) \in \mathcal{S}$ `for every` $q \in \mathsf{supp}(\rho_t)$. `*/`

**6** $\quad$ Observe realized posterior mean $q_t \sim \rho_t$ and buyer's purchase decision $a_t \in \{0, 1\}$.

**7** $\quad$ Update $\left\{ \mathcal{N}_{t+1}(x), N_{t+1}(x), \bar{D}_{t+1}(x) \right\}_{x \in \mathcal{S}}$.

---

exists a polynomial (w.r.t. the number of discontinuities) algorithm based on convex programming that can find an optimal advertising strategy (see Proposition 2 in [4]). Thus, an exhaustively search over the discretized price space $\mathcal{P}$ can lead to an optimal solution to the program $\mathsf{P}_t^{\mathsf{UCB}}$. $\qquad\square$

We conclude this section with the following remark on the extension to unbounded type support.

**Remark 2.3.** *Our algorithm and analysis can be extended to the case with unbounded type support (e.g., $\Theta = [0, \infty)$). In particular, since the price is bounded by $[0, U]$, and the quality is bounded within $[0, 1]$, by the monotoncity of the valuation function, we know the critical types $\kappa(p, q)$ induced by any possible $p \in [0, U]$ and $q \in [0, 1]$ is bounded within $[\kappa(0, 1), \kappa(U, 0)]$. Thus, an instance with unbounded type support is equivalent to an instance with bounded type support $[\kappa(0, 1), \kappa(U, 0)]$.*

## 3 Regret Analysis: Proof Overview of Theorem 1.1

In this section, we present our main regret bound for Algorithm 1, as stated in Theorem 1.1. Specifically, we show that for any type CDF $F$, given a valuation function $v(\theta, \omega)$ that is linear in product quality $\omega$ and satisfies Assumption 1, our algorithm (Algorithm 1 with parameter $\varepsilon = \Theta\big( (m \log T / T)^{1/3} \big)$), has an expected regret of $O\big( T^{2/3} (m \log T)^{1/3} \big)$. Importantly, for this result we do not assume any smoothness or Lipschitz properties of distribution $F$.

For this result, we consider arbitrary but discrete quality space $\Omega$ of cardinality $m$. Later in Appendix B, we show improved regret bounds for the case of additive valuations and equally-spaced quality space (see Theorem B.1), and also extend to arbitrary large and continuous quality spaces (see Theorem B.2).

**Proof outline.** Recall that in every round $t$, Algorithm 1 sets the price $p_t$ and advertising strategy $\rho_t$ as an optimal solution of program $\mathsf{P}_t^{\mathsf{UCB}}$ that approximates the benchmark $\mathsf{P}_{\mathsf{OPT}}$ in two ways. Firstly, it restricts the price and support of advertising strategy to be in a discretized space $\mathcal{P} \times \{\mathcal{Q}_p, p \in \mathcal{P}\}$. Secondly, it approximates the true demand function with an upper bound $D_t^{\mathsf{UCB}}$. Our proof consists of two main steps that bound the errors due to each of the above approximations. Due to the space limit, all missing proofs in this section are deferred to Appendix A.

- **Step 1: bounding the discretization error using a rounding argument (see Appendix A.1).**
  To separate the discretization error from the error due to demand function estimation, we consider an intermediate optimization problem $\widetilde{\mathsf{P}}$ (in Appendix A.1) obtained on replacing the UCB demand function $D_t^{\mathsf{UCB}}$ with the true demand function $D$ (while keeping the discretized space for $p, \rho$). Let $\widetilde{p}^*, \widetilde{\rho}^*$ be an optimal solution of program $\widetilde{\mathsf{P}}$. We show that the revenue $\mathsf{Rev}(\widetilde{p}^*, \widetilde{\rho}^*)$ is sufficiently close (within $2\varepsilon$) to the optimal revenue $\mathsf{Rev}(p^*, \rho^*)$. This bound is obtained using a careful *rounding* argument: we show that the optimal price $p^*$ and the optimal advertising $\rho^*$ can be rounded to a new price $p^\dagger$ and a new advertising $\rho^\dagger$ that satisfy
  (i) feasibility (**Lemma A.4**): $p^\dagger \in \mathcal{P}, \mathsf{supp}(\rho^\dagger) \subseteq \mathsf{supp}(\mathcal{Q}_{p^\dagger})$; and

(ii) revenue guarantee (**Lemma A.5**): $\mathsf{Rev}(p^\dagger, \rho^\dagger) \geq \mathsf{Rev}(p^*, \rho^*) - 2\epsilon$.

- **Step 2: bounding estimation error and establishing optimism (see Appendix A.3).**
  Next, we show that the UCB estimates of the demand function $D_t^{\mathsf{UCB}}(x), x \in \mathcal{S}$ converge to the true demand function $D$ with high probability, along with concentration bounds on the gap between the true and estimated function (**Lemma A.7**). This allows us to show that

  (i) Revenue optimism (**Lemma A.8**): we show that the algorithm's revenue estimates are (almost) optimistic, i.e., $\mathsf{Rev}_t^{\mathsf{UCB}}(p_t, \rho_t) \geq \mathsf{Rev}(\widetilde{p}^*, \widetilde{\rho}^*) \geq \mathsf{Rev}(p^*, \rho^*) - 2\varepsilon$.

  (ii) Revenue approximation (**Lemma A.9**): we show that the optimistic revenue estimates are close to the true revenue in round $t$, with the gap between the two being inversely proportional to the number of observations, in particular, $\mathsf{Rev}_t^{\mathsf{UCB}}(p_t, \rho_t) - \mathsf{Rev}(p_t, \rho_t) \leq 5 p_t \mathbb{E}_{q \sim \rho_t}\left[\sqrt{\log T / N_t(\kappa(p_t, q))}\right]$.

Finally, in Appendix A.7 we put it all together to bound the regret as stated in Theorem 1.1. We first use the above observations to show that regret over each round can be roughly bounded as $2\epsilon T + 5 p_t \mathbb{E}_{q \sim \rho_t}\left[\sqrt{\log T / N_t(\kappa(p_t, q))}\right]$. Then, using the constraint $\sum_{x \in \mathcal{S}} N_T(x) \leq T$, we show that in the worst case, total regret over time $T$ is bounded by $O(T\epsilon + \sqrt{|\mathcal{S}| T \log T})$. The theorem is then obtained by substituting $|\mathcal{S}| = O(m/\epsilon)$ and optimizing $\varepsilon$.

## 4 Conclusions and Future Direction

In this work, we use a foundational information design framework, Bayesian persuasion, to model the effect of an advertising strategy on customers' beliefs about product quality and consequently their purchase decisions. Our formulation combines the Bayesian persuasion model with dynamic pricing and learning to quantify the tradeoffs between the design of the pricing and advertising strategies and their combined impact on the revenue outcomes. Without any apriori knowledge of the demand function, we show that there exists an efficient online policy that has the regret $O(T^{2/3}(m \log T)^{1/3})$ for a finite state space with cardinality $m$. This result implies that when the number of the states $m$ is a constant, there is almost no additional learning regret for the seller to additionally learn the optimal advertising compared to the dynamic pricing (without advertising) for non-parametric demand learning problem. There are interesting future directions from this work. First, in our current formulation, buyers' valuation is linear w.r.t the product states. It would be interesting to explore whether our results could be generalized to more general valuation function. Secondly, in our setting, a buyer with i.i.d private type arrives at each time round, and leaves the market no matter what the purchase decision is. However, in practice, buyers may strategize their purchase time for a more favorable price [2, 29, 60]. How to model the advertising effect with strategic buyers and achieve efficient demand learning is another interesting future direction.

## Acknowledgements

We thank the anonymous reviewers for the helpful comments. This work was supported in part by grants G-2020-13917 from Alfred P. Sloan Foundation and NSF CAREER award 1846792.

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
