# A Detailed Discussions and Proofs of Section 3

## A.1 A rounding procedure to bound the discretization error

In this subsection, we bound the loss in revenue due to discretization. Specifically, let $\widetilde{p}^*, \widetilde{\rho}^*$ be the solution of the following program:

$$\max_{p \in \mathcal{P}} \max_{\rho} \quad p \sum_{i \in [m]} \lambda_i \int_0^1 \rho_i(q) D(\kappa(p,q)) dq$$

$$\text{s.t.} \quad \frac{\sum_{i \in [m]} \lambda_i \rho_i(q) \bar{\omega}_i}{\sum_{i \in [m]} \lambda_i \rho_i(q)} = q, \qquad q \in \mathcal{Q}_p \tag{$\widetilde{\mathsf{P}}$}$$

$$\rho_i \in \Delta^{\mathcal{Q}_p}, \qquad i \in [m] \,.$$

The main result of this subsection is then summarized as follows:

**Proposition A.1.** *For any type CDF F, we have* $\mathsf{Rev}(p^*, \rho^*) - \mathsf{Rev}(\widetilde{p}^*, \widetilde{\rho}^*) \leq 2\varepsilon$.

We prove the above result by showing that we can use a rounding procedure (see Procedure 2) to round the optimal price $p^*$ and the optimal advertising $\rho^*$ to a new price $p^\dagger$ and a new advertising $\rho^\dagger$ that satisfy: (i) $p^\dagger \in \mathcal{P}$, $\mathsf{supp}(\rho^\dagger) \subseteq \mathcal{Q}_{p^\dagger}$; and (ii) the revenue loss $\mathsf{Rev}(p^*, \rho^*) - \mathsf{Rev}(p^\dagger, \rho^\dagger) \leq 2\varepsilon$. It is worth noting that Procedure 2, which we believe is of independent of interest, works for any buyer valuation function that is linear in quality and satisfies Assumption 1. And it only uses the knowledge of critical-type function $\kappa(\cdot, \cdot)$ and prior distribution $\lambda$. In particular, Procedure 2 does not depend on any knowledge or estimates about the unknown demand function. Indeed, Proposition A.1 still holds if we replace the demand function $D$ in the revenue formulation (1) with any monotone non-increasing function. A graphic illustration of Procedure 2 is provided in Figure 1.

In this subsection, we provide details of our rounding procedure and its graphical illustration.

---

**Procedure 2:** Rounding$(p, \rho)$: A critical-type guided procedure to round the strategy $p, \rho$

**Input:** $\varepsilon$, a price $p$ such that $p \geq 2\varepsilon$, and an advertising $\rho$ such that $p, \rho$ satisfy Lemma A.2 and Lemma A.3.

**Output:** A price $p^\dagger \in \mathcal{P}$, an advertising $\rho^\dagger$ satisfy $\mathsf{supp}(\rho^\dagger) \subseteq \mathcal{Q}_{p^\dagger}$

1 **Initialization:** Let the set $\mathcal{Q} \leftarrow \emptyset$. `// The set Q contains the support of the`
   `advertising `$\rho^\dagger$`.`

2 Define price $p^\dagger \leftarrow \max\{p' \in \mathcal{P} : p - 2\varepsilon \leq p' \leq p - \varepsilon\}$.

3 **for** *each posterior mean* $q \in \mathsf{supp}(\rho)$ **do**

4     **if** $q \in \mathcal{Q}_{p^\dagger}$ **then** `// Namely, for this case `$\kappa(p^\dagger, q) \in \mathcal{S}$

5         $\mathcal{Q} \leftarrow \mathcal{Q} \cup \{q\}$, and let $\rho^\dagger(q) = \rho(q)$, and let
           $\{i' \in [m] : \rho_{i'}^\dagger(q) > 0\} = \{i' \in [m] : \rho_{i'}(q) > 0\}$.

6     **else**

7         Suppose $\{i' \in [m] : \rho_{i'}(q) > 0\} = \{i, j\}$ where $i < j$.

8         Let $x \triangleq \kappa(p, q)$, and let $x^\dagger \triangleq \kappa(p^\dagger, q) \in ((z-1)\varepsilon, z\varepsilon)$ for some $z \in \mathbb{N}^+$.

9         Let $q_L, q_R$ satisfy $\kappa(p^\dagger, q_L) = z\varepsilon$, $\kappa(p^\dagger, q_R) = (z-1)\varepsilon$.

10       Let $q_L^\dagger \triangleq q_L \vee \bar{\omega}_i$, and let $q_R^\dagger \triangleq q_R \wedge \bar{\omega}_j$.

11       $\mathcal{Q} \leftarrow \mathcal{Q} \cup \{q_L^\dagger, q_R^\dagger\}$.
         `/* The conditional probabilities below are constructed to satisfy (`BC`).`
         `*/`

12       Let $\rho_i^\dagger(q_L^\dagger) = \frac{\bar{\omega}_j - q_L^\dagger}{\bar{\omega}_j - \bar{\omega}_i} \frac{1}{\lambda_i} \frac{\rho(q)(q_R^\dagger - q)}{q_R^\dagger - q_L^\dagger}$ and $\rho_i^\dagger(q_R^\dagger) = \rho_i(q) - \rho_i^\dagger(q_L^\dagger)$;
         $\rho_j^\dagger(q_L^\dagger) = \frac{q_L^\dagger - \bar{\omega}_i}{\bar{\omega}_j - \bar{\omega}_i} \frac{1}{\lambda_j} \frac{\rho(q)(q_R^\dagger - q)}{q_R^\dagger - q_L^\dagger}$ and $\rho_j^\dagger(q_R^\dagger) = \rho_j(q) - \rho_j^\dagger(q_L^\dagger)$.

---

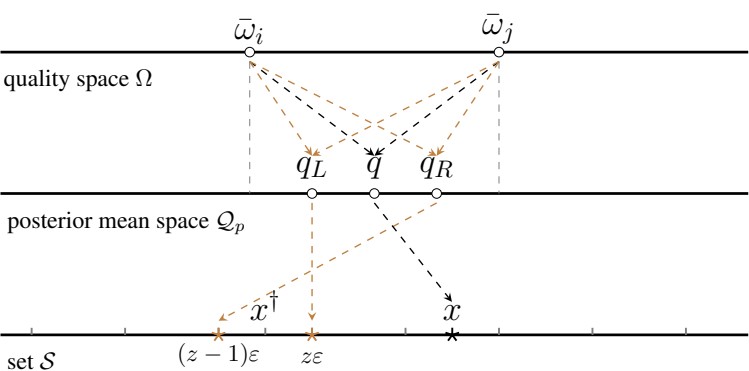

Figure 1: Graphical illustration for Procedure 2. Given the input price and advertising $(p, \rho)$, fix a posterior mean $q \in \mathsf{supp}(\rho)$ where $\{i' \in [m] : \rho_{i'}(q) > 0\} = \{i, j\}$ (drawn in black dashed line). According to the procedure, we first identify $x = \kappa(p, q)$, and $x^\dagger = \kappa(p^\dagger, q) \in ((z-1)\varepsilon, z\varepsilon)$ where the constructed price $p^\dagger$ is defined as in the procedure. We then find two posterior means $q_L, q_R$ (here $q_L \geq \bar{\omega}_i, q_R \leq \bar{\omega}_j$) such that $\kappa(p^\dagger, q_L) = z\varepsilon$ and $\kappa(p^\dagger, q_R) = (z-1)\varepsilon$ (drawn in brown dashed line), and $\kappa(p^\dagger, q_R) < \kappa(p^\dagger, q_L) < \kappa(p, q)$.

**Details and guarantees of Procedure 2.** Procedure 2 takes as input an input advertising strategy $\rho$ that satisfies $|\{i' \in [m] : \rho_{i'}(q) > 0\}| \leq 2$ for any posterior mean $q \in \mathsf{supp}(\rho)$. This structural requirement says that in advertising $\rho$, the realized signal either fully reveals the product quality, or randomizes buyer's uncertainty within two product qualities. Indeed, we can show that there exists an optimal advertising strategy for program $\boxed{\mathsf{P}_{\mathsf{OPT}}}$ that satisfies this structural requirement:

**Lemma A.2** (see, e.g., [33]). *There exists an optimal advertising strategy $\rho^*$ satisfying that $|\{i \in [m] : \rho_i^*(q) > 0\}| \leq 2$ for every $q \in \mathsf{supp}(\rho^*)$.*

Intuitively, the above result is an implication of the fact that the extreme points of the distributions with fixed expectations are binary-supported distributions. Meanwhile, we can also deduce the following property for the optimal price $p^*$ and optimal advertising $\rho^*$:

**Lemma A.3.** *There exist an optimal price $p^*$ and optimal advertising $\rho^*$ such that for any posterior mean $q \in \mathsf{supp}(\rho^*)$ and $q \notin \Omega$, we have that $p^* \leq \max_{\theta \in \Theta} v(\theta, q)$.*

The above property follows from the observation that if there exists a posterior mean $q \in \mathsf{supp}(\rho^*)$ and $q \notin \Omega$, then from Lemma A.2, it must be the case $\{i' \in [m] : \rho_{i'}^*(q) > 0\} = \{i, j\}$ for some $i < j$ such that $\bar{\omega}_i < q < \bar{\omega}_j$. Now, if $p^* > \max_{\theta \in \Theta} v(\theta, q)$, then for all types of buyers, the valuation at this posterior mean is below the given price $p^*$ so that this posterior mean does not contribute to the revenue; therefore one can decompose the probability over this posterior mean $\rho^*(q)$ to probabilities over $\bar{\omega}_i, \bar{\omega}_j$ without losing any revenue and thus obtain a $p^*, \rho^*$ with the desired property.

*Proof of Lemma A.3.* Let us fix the optimal price $p^*$ and optimal advertising $\rho^*$. Suppose there exists a posterior mean $q \in \mathsf{supp}(\rho^*)$ and $q \notin \Omega$, then from Lemma A.2, it must be the case $\{i' \in [m] : \rho_{i'}^*(q) > 0\} = \{i, j\}$ for some $i < j$ that $\bar{\omega}_i < q < \bar{\omega}_j$. Suppose $p^* > \max_{\theta \in \Theta} \kappa(\theta, q)$, then it is easy to see that the revenue contributed from this posterior mean $p^* \sum_i \lambda_i \rho_i^*(q) D(\kappa(p^*, q)) = 0$. Thus, decoupling this posterior mean $q$ to the states $\bar{\omega}_i$ and $\bar{\omega}_j$ will not lose any revenue. $\qquad\square$

With the above Lemma A.2 and Lemma A.3, we now formally present two guarantees on the price and advertising strategy obtained from Procedure 2.

**Lemma A.4** (Feasibility guarantee). *Given an input price and advertising strategy $p, \rho$ satisfying the properties stated in Lemma A.2 and Lemma A.3, the output price $p^\dagger$ and the advertising strategy $\rho^\dagger$ from Procedure 2 satisfies: $p^\dagger \in \mathcal{P}$, $\rho^\dagger$ is a feasible advertising and satisfies $\kappa(p^\dagger, q) \in \mathcal{S}$ for every $q \in \mathsf{supp}(\rho^\dagger)$.*

**Lemma A.5** (Revenue guarantee). *Fix a price $p \geq 2\varepsilon$ and a feasible advertising strategy $\rho$, let $p^\dagger, \rho^\dagger = \mathsf{Rounding}(p, \rho)$ be the output from Procedure 2, then we have $\mathsf{Rev}(p, \rho) - \mathsf{Rev}(p^\dagger, \rho^\dagger) \leq 2\varepsilon$.*

The proofs of the above two lemmas are provided in Appendix A.2. With these two guarantees, we can now prove Proposition A.1:

*Proof of Proposition A.1.* Let $p^\dagger, \rho^\dagger = \mathsf{Rounding}(p^*, \rho^*)$, then we have $\mathsf{Rev}(p^*, \rho^*) - \mathsf{Rev}(\widetilde{p}^*, \widetilde{\rho}^*) \leq \mathsf{Rev}(p^*, \rho^*) - \mathsf{Rev}(p^\dagger, \rho^\dagger) \leq 2\varepsilon$ where the first inequality follows from the feasibility guarantee of price $p^\dagger$ and advertising $\rho^\dagger$ in Lemma A.4 and the definition of $\widetilde{p}^*, \widetilde{\rho}^*$, and the second inequality follows from revenue guarantee in Lemma A.5. $\qquad\square$

## A.2 Proofs of Lemma A.4 and Lemma A.5

In this subsection, we provide proofs of Lemma A.4 and Lemma A.5. At a high-level the argument is as follows. Given an input price $p$, Procedure 2 outputs the closest price $p^\dagger \in \mathcal{P}$ that satisfies $p - 2\varepsilon \leq p^\dagger \leq p - \varepsilon$. Given an input advertising $\rho$, for every posterior mean $q \in \mathsf{supp}(\rho)$ and $q \notin \mathcal{Q}_{p^\dagger}$, with Lemma A.2, there must exist two qualities $\bar{\omega}_i, \bar{\omega}_j$ where $i < j$ such that $\{i' \in [m] : \rho_{i'}(q) > 0\} = \{i, j\}$. For such posterior mean $q$, Procedure 2 first identifies the critical type $x = \kappa(p, q)$ and $x^\dagger = \kappa(p^\dagger, q)$ where $x^\dagger$ lies within a grid $((z - 1)\varepsilon, z\varepsilon)$ for some $z \in \mathbb{N}^+$. Then, Procedure 2 utilizes the critical-type function $\kappa(p^\dagger, \cdot)$ for the constructed price $p^\dagger$ to find two posterior means $q_L, q_R$ such that they satisfy: $\kappa(p^\dagger, q_L) = z\varepsilon$ and $\kappa(p^\dagger, q_R) = (z - 1)\varepsilon$. [6] To construct a feasible advertising strategy, we then round $q_L$ up to be $\bar{\omega}_i$ when $q_L < \bar{\omega}_i$ happens and round $q_R$ down to be $\bar{\omega}_j$ when $q_R > \bar{\omega}_j$ happens. By Assumption 1, and together with Lemma A.3, we can also show that the constructed two posterior means $q_L, q_R$ satisfy (a): $q_L < q < q_R$; and moreover (b): $\kappa(p^\dagger, q_L) \leq \kappa(p, q), \kappa(p^\dagger, q_R) \leq \kappa(p, q)$. The relation (a) enables us to decompose the probability over this posterior mean $\rho(q)$ into probabilities over the two posterior means $q_L, q_R$ while still satisfying (BC) condition. Together with the monotonicity of demand function $D$, the relation (b) can guarantee that the revenue of the output from Procedure 2 is $2\varepsilon$-approximate of the revenue of the input.

**Lemma A.4** (Feasibility guarantee). *Given an input price and advertising strategy $p, \rho$ satisfying the properties stated in Lemma A.2 and Lemma A.3, the output price $p^\dagger$ and the advertising strategy $\rho^\dagger$ from Procedure 2 satisfies: $p^\dagger \in \mathcal{P}$, $\rho^\dagger$ is a feasible advertising and satisfies $\kappa(p^\dagger, q) \in \mathcal{S}$ for every $q \in \mathsf{supp}(\rho^\dagger)$.*

*Proof.* $p^\dagger \in \mathcal{P}$ holds trivially by construction. In below, we first show that the output $\rho^\dagger$ is indeed a feasible advertising strategy, and then prove that $\kappa(p^\dagger, q') \in \mathcal{S}$ for every $q' \in \mathsf{supp}(\rho^\dagger)$. In below analysis, let the price $p$ and the advertising strategy $\rho$ be the input of Procedure 2, and we will focus on an arbitrary posterior mean $q \in \mathsf{supp}(\rho)$ and analyze the corresponding construction for $\rho^\dagger$ from the posterior mean $q$.

$\rho^\dagger$ **as a feasible advertising strategy:** Clearly, a strategy $\rho^\dagger$ is a feasible advertising strategy must satisfy that $\rho^\dagger \in \Delta([0, 1])$, i.e., $\rho^\dagger$ is indeed a distribution over $[0, 1]$; and the associated conditional distributions $(\rho_i^\dagger)_{i \in [m]}$ must be Bayes-consistent as defined in (BC). In below analysis, by Lemma A.3, we assume that $p^* \leq v(1, q)$ for every $q \in \mathsf{supp}(\rho^*)$ and $q \notin \Omega$.

We first prove that the constructed advertising strategy $\rho^\dagger$ is indeed a feasible distribution. We focus on the case where $q \notin \mathcal{Q}_{p^\dagger}$. In this case, we must have $p^\dagger \in (v(0, q), v(1, q))$, otherwise it either $p^\dagger \leq v(0, q)$ or $p^\dagger = v(1, q)$ which both cases falls into the scenario $q \in \mathcal{Q}_{p^\dagger}$. We first show the following claim: For any posterior mean $q \in \mathsf{supp}(\rho)$ with $\{i' \in [m] : \rho_{i'}(q) > 0\} = \{i, j\}$, we have $q_L^\dagger \leq q \leq q_R^\dagger$. To see this, by definition, we have $v(x^\dagger, q) = p^\dagger, v(z\varepsilon, q_L) = p^\dagger, v((z - 1)\varepsilon, q_R) = p^\dagger$, where $x^\dagger \in ((z - 1)\varepsilon, z\varepsilon)$. By Assumption 1 where buyer's valuation $v(\cdot, \cdot)$ is monotone non-decreasing, we know $q_L \leq q \leq q_R$. Now we show that $\rho_i^\dagger(q_L^\dagger) \leq \rho_i(q)$ (similar analysis can also show that $\rho_j^\dagger(q_L^\dagger) \leq \rho_j(q)$). To see this, notice that from Lemma A.2, we must

---

[6]To see that such $q_L$ and $q_R$ always exist, note that because the valuation function is assumed to be monotone increasing in type $\theta$ (see Assumption 1b), given any $\theta, p, q$, if we have $v(\theta, q) = p$, then $\kappa(p, q) = \theta$. Therefore, $q_L$ and $q_R$ are the values of $q$ satisfying $v(z\epsilon, q) = p^\dagger$ and $v((z - 1)\epsilon, q) = p^\dagger$, respectively. Now, under linearity in quality, $v(\theta, q)$ is continuous in $q$ for any given $\theta$, which means that such solutions $q_L$ and $q_R$ always exist.

have

$$\rho(q) = \lambda_i \rho_i(q) + \lambda_j \rho_j(q), \quad \frac{\lambda_i \rho_i(q)\bar{\omega}_i + \lambda_j \rho_j(q)\bar{\omega}_j}{\rho(q)} = q \, .$$

Thus, we must have $\rho_i(q) = \frac{\rho(q) \cdot (\bar{\omega}_j - q)}{\lambda_i (\bar{\omega}_j - \bar{\omega}_i)}$. Hence,

$$
\begin{aligned}
\rho_i(q) - \rho_i^\dagger(q_L^\dagger) &= \frac{\rho(q) \cdot (\bar{\omega}_j - q)}{\lambda_i (\bar{\omega}_j - \bar{\omega}_i)} - \frac{\bar{\omega}_j - q_L^\dagger}{\bar{\omega}_j - \bar{\omega}_i} \cdot \frac{1}{\lambda_i} \cdot \frac{\rho(q) \cdot (q_R^\dagger - q)}{q_R^\dagger - q_L^\dagger} \\
&= \frac{\rho(q) \cdot (\bar{\omega}_j - q_L^\dagger)}{\lambda_i (\bar{\omega}_j - \bar{\omega}_i)} \cdot \left( \frac{\bar{\omega}_j - q}{\bar{\omega}_j - q_L^\dagger} - \frac{\rho(q) \cdot (q_R^\dagger - q)}{q_R^\dagger - q_L^\dagger} \right) \geq 0 \, ,
\end{aligned}
$$

where the last inequality follows from the fact that $\bar{\omega}_i \leq q_L^\dagger \leq q \leq q_R^\dagger \leq \bar{\omega}_j$. Together with the fact that $\rho_i(q) \leq 1$, this shows that value $\rho_i^\dagger(q_L^\dagger) \in [0, 1]$.

We now argue that in the constructed advertising strategy $\rho^\dagger$, the summation of all conditional probabilities for realizing all possible posterior mean in $\mathcal{Q}$ indeed equals to 1. Notice that from Procedure 2, for any posterior mean $q \in \mathsf{supp}(\rho)$ with $\{i' \in [m] : \rho_{i'}(q) > 0\} = \{\bar{\omega}_i, \bar{\omega}_j\}$, the constructed advertising strategy $\rho^\dagger$ included two posterior means $q_L^\dagger, q_R^\dagger$, and the probabilities for realizing posterior means $q_L^\dagger, q_R^\dagger$ are $\rho^\dagger(q_L^\dagger) = \lambda_i \rho_i^\dagger(q_L^\dagger) + \lambda_j \rho_j^\dagger(q_R^\dagger)$ (resp. $\rho^\dagger(q_R^\dagger) = \lambda_i \rho_i^\dagger(q_R^\dagger) + \lambda_j \rho_j^\dagger(q_R^\dagger)$). By construction, we have $\rho^\dagger(q_L^\dagger) + \rho^\dagger(q_R^\dagger) = \rho(q)$. Hence, from

$$\sum_{q \in \mathsf{supp}(\rho)} \rho^\dagger(q_L^\dagger) + \rho^\dagger(q_R^\dagger) = \sum_{q \in \mathsf{supp}(\rho)} \rho(q) = 1,$$

we know the constructed advertising strategy $\rho^\dagger$ is indeed a feasible distribution.

We now show that the constructed advertising strategy $\rho^\dagger$ indeed satisfies the condition (BC). In other words, we want to prove that for every $q' \in \mathsf{supp}(\rho^\dagger)$, we have

$$\frac{\sum_{i \in [m]} \lambda_i \rho_i^\dagger(q)\bar{\omega}_i}{\sum_{i \in [m]} \lambda_i \rho_i^\dagger(q)} = q'$$

Notice that when $\{i' \in [m] : \rho_{i'}(q) > 0\} = \{i\}$, the condition (BC) holds trivially. When $\{i' \in [m] : \rho_{i'}(q) > 0\} = \{i, j\}$, Procedure 2 adds two posterior means $q_L^\dagger, q_R^\dagger$ to the support of $\rho^\dagger$. For the posterior mean $q_L^\dagger$:

$$
\begin{aligned}
\frac{\lambda_i \rho_i^\dagger(q_L^\dagger)\bar{\omega}_i + \lambda_j \rho_j^\dagger(q_L^\dagger)\bar{\omega}_j}{\lambda_i \rho_i^\dagger(q_L^\dagger) + \lambda_j \rho_j^\dagger(q_L^\dagger)} &= \frac{\frac{\bar{\omega}_j - q_L^\dagger}{\bar{\omega}_j - \bar{\omega}_i} \cdot \frac{\rho(q) \cdot (q_R^\dagger - q)}{q_R^\dagger - q_L^\dagger} \cdot \bar{\omega}_i + \frac{q_L^\dagger - \bar{\omega}_i}{\bar{\omega}_j - \bar{\omega}_i} \cdot \frac{\rho(q) \cdot (q_R^\dagger - q)}{q_R^\dagger - q_L^\dagger} \cdot \bar{\omega}_j}{\frac{\bar{\omega}_j - q_L^\dagger}{\bar{\omega}_j - \bar{\omega}_i} \cdot \frac{\rho(q) \cdot (q_R^\dagger - q)}{q_R^\dagger - q_L^\dagger} + \frac{q_L^\dagger - \bar{\omega}_i}{\bar{\omega}_j - \bar{\omega}_i} \cdot \frac{\rho(q) \cdot (q_R^\dagger - q)}{q_R^\dagger - q_L^\dagger}} \\
&= \frac{\bar{\omega}_j - q_L^\dagger}{\bar{\omega}_j - \bar{\omega}_i} \cdot \bar{\omega}_i + \frac{q_L^\dagger - \bar{\omega}_i}{\bar{\omega}_j - \bar{\omega}_i} \cdot \bar{\omega}_j = q_L^\dagger
\end{aligned}
$$

On the other hand, we observe

$$
\begin{aligned}
\rho^\dagger(q_L^\dagger)q_L^\dagger + \rho^\dagger(q_R^\dagger)q_R^\dagger &= (\lambda_i \rho_i^\dagger(q_L^\dagger) + \lambda_j \rho_j^\dagger(q_L^\dagger))q_L^\dagger + (\lambda_i(\rho_i(q) - \rho_i^\dagger(q_L^\dagger)) + \lambda_j(\rho_j(q) - \rho_j^\dagger(q_L^\dagger)))q_R^\dagger \\
&= (\lambda_i \rho_i(q) + \lambda_j \rho_j(q))q_R^\dagger - (\lambda_i \rho_i^\dagger(q_L^\dagger) + \lambda_j \rho_j^\dagger(q_L^\dagger))(q_R^\dagger - q_L^\dagger) \\
&= \rho(q)q_R^\dagger - \left( \frac{\bar{\omega}_j - q_L^\dagger}{\bar{\omega}_j - \bar{\omega}_i} \cdot \frac{\rho(q) \cdot (q_R^\dagger - q)}{q_R^\dagger - q_L^\dagger} + \frac{q_L^\dagger - \bar{\omega}_i}{\bar{\omega}_j - \bar{\omega}_i} \cdot \frac{\rho(q) \cdot (q_R^\dagger - q)}{q_R^\dagger - q_L^\dagger} \right)(q_R^\dagger - q_L^\dagger) \\
&= \rho(q)q_R^\dagger + \rho(q)(q_R^\dagger - q) = \rho(q)q \, .
\end{aligned}
$$

Thus, for the posterior mean $q_R^\dagger$, we have

$$\frac{\lambda_i \rho_i^\dagger(q_R^\dagger)\bar{\omega}_i + \lambda_j \rho_j^\dagger(q_R^\dagger)\bar{\omega}_j}{\lambda_i \rho_i^\dagger(q_R^\dagger) + \lambda_j \rho_j^\dagger(q_R^\dagger)} = \frac{\lambda_i(\rho_i(q) - \rho_i^\dagger(q_L^\dagger))\bar{\omega}_i + \lambda_j(\rho_j(q) - \rho_j^\dagger(q_L^\dagger))\bar{\omega}_j}{\lambda_i(\rho_i(q) - \rho_i^\dagger(q_L^\dagger)) + \lambda_j(\rho_j(q) - \rho_j^\dagger(q_L^\dagger))}$$

$$= \frac{\lambda_i \rho_i(q)\bar{\omega}_i + \lambda_j \rho_j(q)\bar{\omega}_j - q_L^\dagger \cdot (\lambda_i \rho_i^\dagger(q_L^\dagger) + \lambda_j \rho_j^\dagger(q_L^\dagger))}{\rho(q) - (\lambda_i \rho_i^\dagger(q_L^\dagger) + \lambda_j \rho_j^\dagger(q_L^\dagger))}$$

$$= \frac{\rho(q)q - \rho^\dagger(q_L^\dagger)q_L^\dagger}{\rho(q) - \rho^\dagger(q_L^\dagger)} = \frac{\rho^\dagger(q_R^\dagger)q_R^\dagger}{\rho^\dagger(q_R^\dagger)} = q_R^\dagger \ .$$

We now have shown that the constructed advertising strategy $\rho^\dagger$ indeed satisfies condition (BC).

$\kappa(p^\dagger, q') \in \mathcal{S}$ **for every** $q' \in \mathsf{supp}(\rho^\dagger)$**:** Fix a posterior mean $q \in \mathsf{supp}(\rho)$, we focus on the case $q \notin \mathcal{Q}_{p^\dagger}$ (the other case is trivial by construction), we know that in Procedure 2, the corresponding posterior means $q_L^\dagger, q_R^\dagger \in \mathsf{supp}(\rho^\dagger)$. And either $q_L^\dagger = q_L$ or $q_L^\dagger = \bar{\omega}_i$, either $q_R^\dagger = q_R$ or $q_R^\dagger = \bar{\omega}_j$. When $q_L^\dagger = q_L$, we have $\kappa(p^\dagger, q_L^\dagger) = \kappa(p^\dagger, q_L) = z\varepsilon \in \mathcal{S}$. When $q_L^\dagger = \bar{\omega}_i$, we have $\kappa(p^\dagger, q_L^\dagger) = \kappa(p^\dagger, \bar{\omega}_i) \in \mathcal{S}$ as $p^\dagger \in \mathcal{P}$. Similar analysis also shows that $\kappa(p^\dagger, q_R^\dagger) \in \mathcal{S}$. The proof completes.

$\square$

**Lemma A.5** (Revenue guarantee)**.** *Fix a price $p \geq 2\varepsilon$ and a feasible advertising strategy $\rho$, let $p^\dagger, \rho^\dagger = \mathsf{Rounding}(p, \rho)$ be the output from Procedure 2, then we have $\mathsf{Rev}(p, \rho) - \mathsf{Rev}(p^\dagger, \rho^\dagger) \leq 2\varepsilon$.*

*Proof.* We provide the proof when the input to Procedure 2 is $p^*, \rho^*$. The proof only utilizes the monotoncity of the function $D$. In below analysis, let $p^\dagger, \rho^\dagger$ be the advertising strategy output from Procedure 2 with the input $p^*, \rho^*$. We now fix a posterior mean $q \in \mathsf{supp}(\rho^*)$ and consider the following two cases:

**Case 1:** $q \in \mathcal{Q}_{p^\dagger}$. In this case, we have $\kappa(p^\dagger, q) \leq \kappa(p^*, q)$.

**Case 2:** $q \notin \mathcal{Q}_{p^\dagger}$. Let $\{i' \in [m] : \rho_{i'}^*(q) > 0\} = \{i, j\}$. Let $q_L^\dagger, q_R^\dagger$ be the corresponding counterpart in the new advertising strategy $\rho^\dagger$, we now show the following claim:

**Claim A.6.** $\kappa(p^\dagger, q_R^\dagger) \leq \kappa(p^\dagger, q_L^\dagger) \leq \kappa(p^*, q)$.

To see the above claim, recall that in construction, we have $v(x, q) \leq p^*$, $v(x^\dagger, q) = p^\dagger$, and by construction, we have $p^\dagger \leq p^* - \varepsilon$. Thus, by Assumption 1b, we have $\varepsilon \leq p^* - p^\dagger \leq v(x, q) - v(x^\dagger, q) \leq x - x^\dagger$, which implies that $z\varepsilon \leq x^\dagger + \varepsilon \leq x$. Fix any price $p$, from Assumption 1a, we know that the function $\kappa(p, \cdot)$ is monotone non-increasing. Recall that in previous analysis, we have shown $q_L \leq q_L^\dagger \leq q \leq q_R^\dagger \leq q_R$. We thus have $\kappa(p^\dagger, q_R^\dagger) \leq \kappa(p^\dagger, q_L^\dagger) \leq \kappa(p^\dagger, q_L) = z\varepsilon \leq x = \kappa(p^*, q)$.

With the above observations, we have

$$\mathsf{Rev}(p^*, \rho^*) - \mathsf{Rev}(p^\dagger, \rho^\dagger)$$

$$= p^* \sum_{i \in [m]} \lambda_i \int_0^1 \rho_i^*(q) D(\kappa(p^*, q)) dq - p^\dagger \sum_{i \in [m]} \lambda_i \int_0^1 \rho_i^\dagger(q) D(\kappa(p^\dagger, q)) dq$$

$$= p^* \int_0^1 \rho^*(q) D(\kappa(p^*, q)) dq - p^\dagger \int_0^1 \rho^\dagger(q) D(\kappa(p^\dagger, q)) dq$$

$$\overset{(a)}{\leq} p^* \int_0^1 \rho^*(q) D(\kappa(p^*, q)) dq - (p^* - 2\varepsilon) \int_0^1 \rho^\dagger(q) D(\kappa(p^\dagger, q)) dq$$

$$\overset{(b)}{=} p^* \sum_{q \in \mathsf{supp}(\rho^*)} \rho^*(q) D(\kappa(p^*, q)) - p^* \sum_{q \in \mathsf{supp}(\rho^*)} \left( \rho^\dagger(q_L^\dagger) D(p^\dagger, q_L^\dagger) + \rho^\dagger(q_R^\dagger) D(p^\dagger, q_R^\dagger) \right) + 2\varepsilon$$

$$\overset{(c)}{=} p^* \sum_{q \in \mathsf{supp}(\rho^*)} \left( \rho^*(q) D\left(\kappa(p^*, q)\right) - \left( \rho^\dagger(q_L^\dagger) D\left(\kappa(p^\dagger, q_L^\dagger)\right) + \rho^\dagger(q_R^\dagger) D\left(\kappa(p^\dagger, q_R^\dagger)\right) \right) \right) + 2\varepsilon$$

$$\overset{(d)}{\leq} \sum_{q \in \mathsf{supp}(\rho^*)} p^* \left( \rho^*(q) D(\kappa(p^*, q)) - \left( \rho^\dagger(q_L^\dagger) D(\kappa(p^*, q)) + \rho^\dagger(q_R^\dagger) D(\kappa(p^*, q)) \right) \right) + 2\varepsilon$$

$$\overset{(e)}{=} \sum_{q \in \mathsf{supp}(\rho^*)} p^* \left( \rho^*(q) D(\kappa(p^*, q)) - \rho^*(q) D(\kappa(p^*, q)) \right) + 2\varepsilon = 2\varepsilon \,,$$

where inequality (a) holds since $p^\dagger \geq p^* - 2\varepsilon$; in equality (b), we, for simplicity, focus on **else** case, the analysis for other scenarios is the same; equality (c) holds by the construction of the strategy $\rho^\dagger$, inequality (d) holds from Claim A.6; inequality (e) holds since by construction, we have for any $q \in \mathsf{supp}(\rho^\dagger)$, we have $\rho^\dagger(q_L^\dagger) + \rho^\dagger(q_R^\dagger) = \rho^*(q)$. $\qquad \square$

### A.3 Estimation error and optimism

We begin our estimation error analysis by showing that $D_t^{\mathsf{UCB}}(x)$ provides an upper confidence bound on the true demand function $D(x)$ for all $x \in \mathcal{S}$, and deriving a bound on how large it can be compared to $D(x)$.

**Lemma A.7.** *For every $t \geq |\mathcal{S}| + 1$, the following holds with probability at least $1 - 1/T^2$:*

$$D_t^{\mathsf{UCB}}(x) \geq D(x), \qquad\qquad\qquad\qquad \forall x \in \mathcal{S}; \qquad (4)$$

$$D_t^{\mathsf{UCB}}(x) - D(x) \leq 2\sqrt{\frac{16 \log T}{N_t(x)}} + \frac{2\sqrt{(1 + N_t(x)) \ln(1 + N_t(x))}}{N_t(x)}, \quad \forall x \in \mathcal{S}. \qquad (5)$$

To prove the inequalities for the points $x \in \mathcal{S}$, we first show that the empirical estimates $\bar{D}_t(x), \forall x \in \mathcal{S}$ concentrate around the true demand value $D(x)$ as $N_t(x)$ increases. We prove this concentration bound by using a uniform bound given by scalar-valued version of self-normalized martingale tail inequality [1]. The proof of the above lemma is provided in Appendix A.4.

We now analyze how close the seller's optimistic revenue estimates using the upper confidence bound $D_t^{\mathsf{UCB}}$ is to the true revenue. In particular, we have the following result.

**Lemma A.8.** *For every time $t \geq |\mathcal{S}| + 1$, with probability at least $1 - 2/T^2$, we have*

$$\mathsf{Rev}(\widetilde{p}^*, \widetilde{\rho}^*) \leq \mathsf{Rev}_t^{\mathsf{UCB}}(\widetilde{p}^*, \widetilde{\rho}^*) \leq \mathsf{Rev}_t^{\mathsf{UCB}}(p_t, \rho_t)$$

The above results follow from the bounds in Lemma A.7 where we established that $D_t^{\mathsf{UCB}}(x) \geq D(x)$ with high probability. The proof of the above Lemma A.8 is provided in Appendix A.5.

Next we show that we can also upper bound $\mathsf{Rev}_t^{\mathsf{UCB}}(p_t, p_t) - \mathsf{Rev}(p_t, \rho_t)$ by applying the results in Lemma A.7 again.

**Lemma A.9.** *For every time $t \geq |\mathcal{S}| + 1$, with probability at least $1 - 2/T^2$, we have*

$$\mathsf{Rev}_t^{\mathsf{UCB}}(p_t, \rho_t) - \mathsf{Rev}(p_t, \rho_t) \leq 5p_t \sum_{q \in \mathsf{supp}(\rho_t)} \rho_t(q) \sqrt{\frac{\log T}{N_t(\kappa(p_t, q))}}$$

The proof of above Lemma A.9 is provided in Appendix A.6. Intuitively, the difference between the estimated seller's revenue $\mathsf{Rev}_t^{\mathsf{UCB}}(p_t, p_t)$, and the true expected revenue $\mathsf{Rev}(p_t, \rho_t)$, can be bounded by a weighted sum (weighted by probabilities $\rho_t(q), q \in \mathsf{supp}(\rho_t)$) of errors in demand estimates on the points $\kappa(p_t, q)$ for $q \in \mathsf{supp}(\rho_t)$: $|D_t^{\mathsf{UCB}}(\kappa(p_t, q)) - D(\kappa(p_t, q))|$.

## A.4 Proof of Lemma A.7

We use the following self-normalized martingale tail inequality to prove the high-probability bounds. In particular, we use the following results obtained in Abbasi-Yadkori et al. [1]:

**Lemma A.10** (Uniform Bound for self-normalized bound for martingales, see [1])**.** *Let $\{\mathcal{F}_t\}_{t=1}^\infty$ be a filtration. Let $\{Z_t\}_{t=1}^\infty$ be a sequence of real-valued variables such that $Z_t$ is $\mathcal{F}_t$-measurable. Let $\{\eta_t\}_{t=1}^\infty$ be a sequence of real-valued random variables such that $\eta_t$ is $\mathcal{F}_{t+1}$-measurable and is conditionally $R$-sub-Gaussian. Let $V > 0$ be deterministic. For any $\delta > 0$, with probability at least $1 - \delta$, for all $t \geq 0$:*

$$\left| \sum_{s=1}^t \eta_s Z_s \right| \leq R \sqrt{2 \left( V + \sum_{s=1}^t Z_s^2 \right) \ln \left( \frac{\sqrt{V + \sum_{s=1}^t Z_s^2}}{\delta \sqrt{V}} \right)} \tag{6}$$

**Lemma A.7.** *For every $t \geq |\mathcal{S}| + 1$, the following holds with probability at least $1 - 1/T^2$:*

$$D_t^{\mathsf{UCB}}(x) \geq D(x), \qquad\qquad\qquad \forall x \in \mathcal{S}; \tag{4}$$

$$D_t^{\mathsf{UCB}}(x) - D(x) \leq 2\sqrt{\frac{16 \log T}{N_t(x)}} + \frac{2\sqrt{(1 + N_t(x)) \ln(1 + N_t(x))}}{N_t(x)}, \quad \forall x \in \mathcal{S}. \tag{5}$$

*Proof.* To prove the results, we first show the following concentration inequality for the empirical demand estimates of the points in the set $\mathcal{S}$: the following holds with probability at least $1 - 1/T^2$,

$$\left| \bar{D}_t(x) - D(x) \right| \leq \sqrt{\frac{16 \log T}{N_t(x)}} + \frac{\sqrt{(1 + N_t(x)) \ln(1 + N_t(x))}}{N_t(x)}, \quad \forall x \in \mathcal{S}. \tag{7}$$

To prove the above inequality, we fix an arbitrary $x \in \mathcal{S}$. We define the random variable $Z_t(x) = \mathbf{1}[\kappa(p_t, q_t) = x]$, We also define random variable $\eta_t(x) = a_t(x) - D(x)$ if $Z_t(x) = 1$ at time step $t$. Then by definition, we know that the sequence $\{\sum_{s=1}^t \eta_t(x) Z_t(x)\}$ is a martingale adapted to $\{\mathcal{F}_{t+1}\}_{t=0}^\infty$. Moreover, the sequence of the variable $\{Z_t(x)\}_{t=1}^\infty$ is $\mathcal{F}_t$-measurable, and the variable $\eta_t(x)$ is 1-sub-Gaussian. Now take $V = 1$ and substitute for $\eta_t(x) = a_t(x) - D(x)$, apply the uniform bound obtained in Lemma A.10, we have for any $t \geq |\mathcal{S}| + 1$, the following holds with probability at least $1 - \delta$,

$$\left| \sum_{s=1}^t (a_s(x) - D(x)) Z_s(x) \right| \leq \sqrt{2 \left( 1 + \sum_{s=1}^t Z_s(x)^2 \right) \ln \left( \frac{\sqrt{1 + \sum_{s=1}^t Z_s(x)^2}}{\delta} \right)}$$

Observe that in the above inequality, the term $\left| \sum_{s=1}^t (a_s(x) - D(x)) Z_s(x) \right|$ is exactly $|\sum_{s \in \mathcal{N}_t(x)} a_s(x) - N_t(x) D(x)|$, and the term $\sum_{s=1}^t Z_s(x)^2$ exactly equals to $N_t(x)$. Dividing both sides with $N_t(x)$, substituting for $\sum_{s=1}^t Z_s(x)^2 = N_t(x)$, we obtain

$$\left| \bar{D}_t(x) - D(x) \right| \leq \frac{1}{N_t(x)} \sqrt{2 (1 + N_t(x)) \ln \left( \frac{\sqrt{1 + N_t(x)}}{\delta} \right)}$$

$$= \sqrt{\frac{2 (1 + N_t(x)) \ln \frac{1}{\delta} + (1 + N_t(x)) \ln (1 + N_t(x))}{N_t(x)^2}}$$

$$\leq \sqrt{\frac{4 \ln \frac{1}{\delta}}{N_t(x)}} + \frac{\sqrt{(1 + N_t(x)) \ln(1 + N_t(x))}}{N_t(x)}$$

where in last inequality we use the fact that $1 + N_t(x) \le 2N_t(x)$, and $\sqrt{u+v} \le \sqrt{u} + \sqrt{v}$ for any $u, v \ge 0$. Setting $\delta = T^{-5}$, we know that the above inequality holds with probability at least $1 - 1/T^5$. Taking the union bound over all choices of $t$ and over all choices of $x \in \mathcal{S}$, we obtain that the first statement holds with probability at least $1 - 1/T^2$ as long as $|\mathcal{S}| \le T$, which is the case for us.

For the inequality (4), for notation simplicity, let $\mathsf{CR}_t(x) \triangleq \sqrt{\frac{16 \log T}{N_t(x)}} + \frac{\sqrt{(1+N_t(x)) \ln(1+N_t(x))}}{N_t(x)}$ be the high-probability error, and we also write $\mathcal{S} = \{x^{(1)}, \ldots, x^{(|\mathcal{S}|)}\}$ where $x^{(i)} < x^{(j)}$ for any $i < j$. Now fix an arbitrary $x^{(i)} \in \mathcal{S}$, fix a time round $t \ge |\mathcal{S}| + 1$. Denote the random variable $i^{\dagger} = \arg\min_{i' : i' \le i} \bar{D}_t(x^{(i')}) + \mathsf{CR}_t(x^{(i')}) \wedge 1$.

$$
\mathbb{P}\left[ D_t^{\mathsf{UCB}}(x^{(i)}) \ge D(x^{(i)}) \right] = 1 - \sum_{j=1}^{i} \mathbb{P}\left[ i^{\dagger} = j \right] \mathbb{P}\left[ D_t^{\mathsf{UCB}}(x^{(i)}) < D(x^{(i)}) \mid i^{\dagger} = j \right]
$$

$$
= 1 - \sum_{j=1}^{i} \mathbb{P}\left[ i^{\dagger} = j \right] \mathbb{P}\left[ \bar{D}_t(x^{(j)}) + \mathsf{CR}_t(x^{(j)}) < D(x^{(i)}) \mid i^{\dagger} = j \right]
$$

$$
\overset{(a)}{\ge} 1 - \sum_{j=1}^{i} \mathbb{P}\left[ i^{\dagger} = j \right] \mathbb{P}\left[ \bar{D}_t(x^{(j)}) + \mathsf{CR}_t(x^{(j)}) < D(x^{(j)}) \mid i^{\dagger} = j \right]
$$

$$
= 1 - \sum_{j=1}^{i} \mathbb{P}\left[ \bar{D}_t(x^{(j)}) + \mathsf{CR}_t(x^{(j)}) < D(x^{(j)}), i^{\dagger} = j \right]
$$

$$
\ge 1 - \sum_{j=1}^{i} \mathbb{P}\left[ \bar{D}_t(x^{(j)}) + \mathsf{CR}_t(x^{(j)}) < D(x^{(j)}) \right]
$$

$$
\overset{(b)}{\ge} 1 - |\mathcal{S}|\delta \ge 1 - T^{-4}
$$

where inequality (a) holds since $D(x^{(j)}) \ge D(x^{(i)})$ for any $j \le i$, and inequality (b) holds follows from earlier analysis where for a fixed $t$ and fixed $x \in \mathcal{S}$, we have $\mathbb{P}\left[ \bar{D}_t(x) + \mathsf{CR}_t(x) < D(x) \right] \le \delta$. Taking the union bound over all choices of $t$ and over all choices of $x \in \mathcal{S}$ finishes the proof.

For the inequality (5), from triangle inequality, we have

$$
\left| D_t^{\mathsf{UCB}}(x) - D(x) \right| \le \left| D_t^{\mathsf{UCB}}(x) - \bar{D}_t(x) \right| + \left| \bar{D}_t(x) - D(x) \right|
$$

$$
\le \sqrt{\frac{16 \log T}{N_t(x)}} + \frac{\sqrt{(1+N_t(x)) \ln(1+N_t(x))}}{N_t(x)} + \left| \bar{D}_t(x) - D(x) \right|
$$

$$
\overset{(a)}{\le} 2\sqrt{\frac{16 \log T}{N_t(x)}} + \frac{2\sqrt{(1+N_t(x)) \ln(1+N_t(x))}}{N_t(x)} ,
$$

where the inequality (a) holds with probability at least $1 - 1/T^2$ according to the first statement we just proved. $\qquad\square$

### A.5   Proof of Lemma A.8

**Lemma A.8.** *For every time $t \ge |\mathcal{S}| + 1$, with probability at least $1 - 2/T^2$, we have*

$$
\mathsf{Rev}(\widetilde{p}^*, \widetilde{\rho}^*) \le \mathsf{Rev}_t^{\mathsf{UCB}}(\widetilde{p}^*, \widetilde{\rho}^*) \le \mathsf{Rev}_t^{\mathsf{UCB}}(p_t, \rho_t)
$$

*Proof.* We begin our analysis by defining the following event. For all $t = |\mathcal{S}| + 1, \ldots, T$, define events $\mathsf{E}_t$

$$
\mathsf{E}_t \triangleq \bigcup_{x \in \mathcal{S}} \left\{ D_t^{\mathsf{UCB}}(x) < D(x) \text{ or } D_t^{\mathsf{UCB}}(x) > D(x) + \sqrt{\frac{16 \log T}{N_t(x)}} + \frac{\sqrt{(1+N_t(x)) \ln(1+N_t(x))}}{N_t(x)} \right\}
$$

From union bound, it follows that

$$\mathbb{P}[\mathsf{E}_t] \leq \sum_{x \in \mathcal{S}} \mathbb{P}\left[D_t^{\mathsf{UCB}}(x) < D(x) \text{ or } D_t^{\mathsf{UCB}}(x) > D(x) + \sqrt{\frac{16 \log T}{N_t(x)}} + \frac{\sqrt{(1 + N_t(x)) \ln(1 + N_t(x))}}{N_t(x)}\right]$$

$$\leq \sum_{x \in \mathcal{S}} \mathbb{P}\left[D_t^{\mathsf{UCB}}(x) < D(x)\right] +$$

$$\sum_{x \in \mathcal{S}} \mathbb{P}\left[D_t^{\mathsf{UCB}}(x) > D(x) + \sqrt{\frac{16 \log T}{N_t(x)}} + \frac{\sqrt{(1 + N_t(x)) \ln(1 + N_t(x))}}{N_t(x)}\right]$$

$$\overset{(a)}{\leq} \frac{2}{T^2}$$

where the inequality (a) follows from inequalities (4) and (5) in Lemma A.7. Recall that whenever $\mathbf{1}[\mathsf{E}_t^{\mathsf{c}}] = 1$, we have

$$\mathsf{Rev}(\widetilde{p}^*, \widetilde{\rho}^*) - \mathsf{Rev}_t^{\mathsf{UCB}}(\widetilde{p}^*, \widetilde{\rho}^*)$$

$$= \widetilde{p}^* \sum_{i \in [m]} \lambda_i \int_0^1 \widetilde{\rho}_i^*(q) D(\kappa(\widetilde{p}^*, q)) dq - \widetilde{p}^* \sum_{i \in [m]} \lambda_i \int_0^1 \widetilde{\rho}_i^*(q) D_t^{\mathsf{UCB}}(\kappa(\widetilde{p}^*, q)) dq$$

$$= \widetilde{p}^* \sum_{i \in [m]} \lambda_i \int_0^1 \widetilde{\rho}_i^*(q) \left(D(\kappa(\widetilde{p}^*, q)) - D_t^{\mathsf{UCB}}(\kappa(\widetilde{p}^*, q))\right) dq \leq 0$$

Thus, whenever $\mathbf{1}[\mathsf{E}_t^{\mathsf{c}}] = 1$, we have

$$\mathsf{Rev}(\widetilde{p}^*, \widetilde{\rho}^*) \leq \mathsf{Rev}_t^{\mathsf{UCB}}(\widetilde{p}^*, \widetilde{\rho}^*) \overset{(a)}{\leq} \mathsf{Rev}(p_t, \rho_t)$$

where inequality (a) follows from our algorithm design. □

## A.6 Proof of Lemma A.9

**Lemma A.9.** *For every time $t \geq |\mathcal{S}| + 1$, with probability at least $1 - 2/T^2$, we have*

$$\mathsf{Rev}_t^{\mathsf{UCB}}(p_t, \rho_t) - \mathsf{Rev}(p_t, \rho_t) \leq 5 p_t \sum_{q \in \mathsf{supp}(\rho_t)} \rho_t(q) \sqrt{\frac{\log T}{N_t(\kappa(p_t, q))}}$$

*Proof.* Follow from the definition of the event $\mathsf{E}_t^{\mathsf{c}}$, when $\mathbf{1}[\mathsf{E}_t^{\mathsf{c}}] = 1$, we have

$$\mathsf{Rev}_t^{\mathsf{UCB}}(p_t, \rho_t) - \mathsf{Rev}(p_t, \rho_t)$$

$$\leq p_t \sum_{i \in [m]} \lambda_i \int_0^1 \rho_{i,t}(q) \left(D_t^{\mathsf{UCB}}(\kappa(p_t, q)) - D(\kappa(p_t, q))\right) dq$$

$$\overset{(a)}{\leq} p_t \sum_{i \in [m]} \lambda_i \int_0^1 \rho_{i,t}(q) \left(\sqrt{\frac{16 \log T}{N_t(\kappa(p_t, q))}} + \frac{\sqrt{(1 + N_t(\kappa(p_t, q))) \ln(1 + N_t(\kappa(p_t, q)))}}{N_t(\kappa(p_t, q))}\right) dq$$

$$\overset{(b)}{\leq} 5 p_t \sum_{i \in [m]} \lambda_i \int_0^1 \rho_{i,t}(q) \sqrt{\frac{\log T}{N_t(\kappa(p_t, q))}} dq$$

$$\overset{(c)}{=} 5 p_t \sum_{q \in \mathsf{supp}(\rho_t)} \rho_t(q) \sqrt{\frac{\log T}{N_t(\kappa(p_t, q))}} \,,$$

where inequality (a) follows from the definition of event $\mathsf{E}_t^{\mathsf{c}}$, inequality (b) follows from the fact that $N_t(x) \leq T, \forall t$ and thus, $\frac{\sqrt{(1+N_t(x)) \ln(1+N_t(x))}}{N_t(x)} \leq \sqrt{\frac{\log T}{N_t(x)}}$, and in last equality (c), we have $\rho_t(q) = \sum_{i \in [m]} \lambda_i \rho_{i,t}(q)$. □

## A.7 Putting it all together

We can now combine the above lemmas to prove Theorem 1.1.

*Proof of Theorem 1.1.* We have that with probability at least $1 - O(1/T)$,

$$
\begin{aligned}
\text{Regret}[T] &\leq |\mathcal{S}| + \mathbb{E}\left[\sum_{t=|\mathcal{S}|+1}^{T} \text{Rev}(p^*, \rho^*) - \text{Rev}(p_t, \rho_t)\right] \\
&\overset{(a)}{\leq} |\mathcal{S}| + 2\varepsilon T + \mathbb{E}\left[\sum_{t=|\mathcal{S}|+1}^{T} \text{Rev}_t^{\text{UCB}}(p_t, \rho_t) - \text{Rev}(p_t, \rho_t)\right] \\
&\overset{(b)}{\leq} |\mathcal{S}| + 2\varepsilon T + 5U\mathbb{E}\left[\sum_{t=|\mathcal{S}|+1}^{T}\sum_{q\in\text{supp}(\rho_t)} \rho_t(q)\sqrt{\frac{\log T}{N_t(\kappa(p_t, q))}}\right] \\
&\overset{(c)}{=} |\mathcal{S}| + 2\varepsilon T + 5U\mathbb{E}\left[\sum_{x\in\mathcal{S}}\sum_{t=|\mathcal{S}|+1}^{T} \beta_t(x)\sqrt{\frac{\log T}{N_t(x)}}\right]\,,
\end{aligned}
\tag{8}
$$

where the first inequality follows from the definition of regret. Inequality (a), follows from Lemma A.8 and Proposition A.1, and inequality (b) follows from Lemma A.9 along with upper bound $U$ on prices $p_t$ in all rounds. For inequality (c), we use that by construction $\kappa(p_t, q) \in \mathcal{S}$ for every posterior mean $q \in \text{supp}(\rho_t)$. and define distribution $\beta_t \in \Delta^{\mathcal{S}}$ over the set $\mathcal{S}$ as

$$
\beta_t(x) = \sum_{q\in\text{supp}(\rho_t):\kappa(p_t, q)=x} \rho_t(q), \quad x \in \mathcal{S}\,.
$$

Define Bernoulli random variable $X_t(x) = \mathbf{1}[x_t = x]$, where $x_t = \kappa(p_t, q_t)$. Then, from the definition of $\beta_t(x)$ observe that $\mathbb{P}[X_t(x) = 1|N_t(x)] = \beta_t(x)$. Also, by definition

$$
N_{t+1}(x) = 1 + \sum_{\ell=|\mathcal{S}|+1}^{t} X_\ell(x) \leq 2N_t(x)\,.
$$

We use these observations below to obtain a bound on the third term in the RHS of (8):

$$
\begin{aligned}
\mathbb{E}\left[\sum_{x\in\mathcal{S}}\sum_{t=|\mathcal{S}|+1}^{T} \beta_t(x)\sqrt{\frac{\log T}{N_t(x)}}\right] &= \mathbb{E}\left[\sum_{x\in\mathcal{S}}\sum_{t=|\mathcal{S}|+1}^{T} \mathbb{E}\left[X_t(x)\sqrt{\frac{\log T}{N_t(x)}}\,\Big|\, N_t(x)\right]\right] \\
&\leq \mathbb{E}\left[\sum_{x\in\mathcal{S}}\sum_{t=|\mathcal{S}|+1}^{T} X_t(x)\sqrt{\frac{2\log T}{N_{t+1}(x)}}\right] \\
&= \mathbb{E}\left[\sum_{x\in\mathcal{S}}\sum_{n=2}^{N_{T+1}(x)} \sqrt{\frac{2\log T}{n}}\right] \\
&\leq \mathbb{E}\left[\sum_{x\in\mathcal{S}} \sqrt{8N_{T+1}(x)\log(T)}\right] \\
&\leq 2\sqrt{2|\mathcal{S}|T\log(T)}\,,
\end{aligned}
$$

Substituting this bound in (8), we obtain that with probability $1 - O(1/T)$,

$$
\text{Regret}[T] \leq |\mathcal{S}| + 2\varepsilon T + 10U\sqrt{2|\mathcal{S}|T\log(T)}\,.
$$

Now, by construction, the set $\mathcal{S}$ has the cardinality of $O(mU/\varepsilon)$. Optimizing $\varepsilon = \Theta((m\log T/T)^{1/3})$ in the above regret bound, we have $\text{Regret}[T] \leq O\left(T^{2/3}(m\log T)^{1/3}\right)$. $\qquad\square$

# B Improved Regret bounds For Additive Valuations

In this section, we discuss improved regret bounds for Algorithm 1 in the case when valuation function is additive, i.e., $v(\theta, \omega) = \theta + \omega$.

First we consider an additional assumption that the product quality domain $\Omega$ is an 'equally-spaced set', which include many natural discrete ordered sets like $\Omega = \{0, 1\}$ or $\Omega = [m]$ that are commonly used in the Bayesian persuasion literature.

**Definition B.1** (Equally-spaced sets). *We say that a discrete ordered set $\Omega = \{\bar{\omega}_1, \ldots, \bar{\omega}_m\}$ is equally-spaced if for all $i \in [m-1]$, $\bar{\omega}_{i+1} - \bar{\omega}_i = c$ for some constant c.*

With this definition, we prove the following improved regret bound.

**Theorem B.1.** *Given an additive valuation function, $v(\theta, \omega) = \theta + \omega$, and equally-spaced product quality domain, $\Omega$ Algorithm 1 with parameter $\varepsilon = \Theta((\log T/T)^{1/3} \wedge 1/m)$ has an expected regret of $O\left(T^{2/3}(\log T)^{1/3} + \sqrt{mT \log T}\right)$.*

Note that a corollary of the above theorem is that the regret is bounded by $O(T^{2/3}(\log T)^{1/3})$ when $m \leq (T/\log T)^{1/3}$ and by $O(\sqrt{mT \log T})$ for larger $m$. The high-level idea behind the above result is as follows. In the previous section (see Section A.7) we show that the expected regret of Algorithm 1 is bounded by $O(T\varepsilon + \sqrt{|\mathcal{S}|T \log T})$. To prove Theorem B.1 we show that in case of additive valuation and the equally-spaced qualities, there exists a discretization parameter $\varepsilon = \Theta((\log T/T)^{1/3} \wedge 1/m)$ such that $\{\kappa(p, \omega)\}_{p \in \mathcal{P}, \omega \in \Omega} \subset \{0, \varepsilon, 2\varepsilon, \ldots, 1\}$. Thus, the constructed set $\mathcal{S}$ satisfies $|\mathcal{S}| = O(m + 1/\varepsilon)$. Substituting the value of $\varepsilon$ then gives the result in Theorem B.1. A formal proof of Theorem B.1 is provided in Appendix C.1.

Furthermore, for additive valuation functions, we can also handle arbitrary large or continuous quality spaces to obtain an $\tilde{O}(T^{3/4})$ regret independent of size of quality space $m$.

**Theorem B.2.** *Given an additive valuation function $v(\theta, \omega) = \theta + \omega$, and arbitrary (discrete or continuous) product quality space $\Omega$, there exists an algorithm (Algorithm 3 in Appendix C.2) that has expected regret of $O(T^{3/4}(\log T)^{1/4})$.*

The proposed Algorithm 3 that achieves the above result is essentially a combination of a pre-processing step and Algorithm 1. In this pre-processing step, we pool the product qualities that are "close enough". This gives us a new problem instance with a smaller discrete product quality space so that we can apply Algorithm 1. With additive valuation function, we show that this reduction does not incur too much loss in revenue. A formal description of the algorithm and proof of Theorem B.2 is provided in Appendix C.2.

# C Missing Proofs of Section B

## C.1 Proof of Theorem B.1

**Theorem B.1.** *Given an additive valuation function, $v(\theta, \omega) = \theta + \omega$, and equally-spaced product quality domain, $\Omega$ Algorithm 1 with parameter $\varepsilon = \Theta((\log T/T)^{1/3} \wedge 1/m)$ has an expected regret of $O\left(T^{2/3}(\log T)^{1/3} + \sqrt{mT \log T}\right)$.*

*Proof.* For additive valuation, we know $\kappa(p, q) = ((p - q) \wedge 1) \vee 0$. Since $q \in [0, 1]$, we know that $\bar{v} = 2, \underline{v} = 0$.

We first prove the regret $O(T^{2/3}(\log T)^{1/3})$ when $m \leq (T/\log T)^{1/3} + 1$. Define the following discretization parameter that will be used to define the discretized price space $\mathcal{P}$ and the discretized type space $\mathcal{S}$ in (2).

$$\varepsilon = \max\left\{\varepsilon' \geq 0 : \frac{1/m}{\varepsilon'} \in \mathbb{N}^+ \wedge \varepsilon' \leq \left(\frac{\log T}{T}\right)^{1/3}\right\} \tag{9}$$

We now argue that the above $\varepsilon = \Theta((\log T/T)^{1/3})$. To see this, let the integers $k_1, k_2 \in \mathbb{N}^+$ satisfy

$$\left\lfloor \frac{1/(m-1)}{(\frac{\log T}{T})^{1/3}} \right\rfloor = k_1, \qquad \left\lfloor \frac{1/(m-1)}{\frac{1}{2}(\frac{\log T}{T})^{1/3}} \right\rfloor = k_2 .$$

By assumption, we have $\frac{1}{m-1} \geq (\log T/T)^{1/3}$, implying $k_1 \geq 1$, and $k_2 \geq 2$. Thus, there must exist an $\varepsilon' \in [(\frac{1}{2}(\log T/T)^{1/3}, (\log T/T)^{1/3}]$ such that $\frac{1/(m-1)}{\varepsilon'} \in [k_1 : k_2]$, which implies that the above defined $\varepsilon = \Theta((\log T/T)^{1/3})$. Suppose $K_\varepsilon \in \mathbb{N}^+$ such that $K_\varepsilon \varepsilon = 1/(m-1)$. By definition of uniformly-spaced qualities, we know that $\bar\omega_i = \frac{i-1}{(m-1)}, \forall i \geq 2$. For a discretized price space $\mathcal{P} = \{\varepsilon, 2\varepsilon, \ldots, 2 - \varepsilon, 2\}$, we know that for any price $p = k_p\varepsilon \in \mathcal{P}$ for some integer $k_p \in \mathbb{N}^+$, we have $\kappa(k_p\varepsilon, \bar\omega_i) = k_p\varepsilon - \bar\omega_i = k_p\varepsilon - (i-1)K_\varepsilon\varepsilon \in \{0, \varepsilon, \ldots, 1\}$. Thus, for the set $\mathcal{S}$ defined in (2) we have $|\mathcal{S}| = O(1/\varepsilon)$. With $\varepsilon$ defined in (9), Algorithm 1 has the desired regret upper bound.

We now prove the regret $O(\sqrt{mT\log T})$ when number of qualities $m > (T/\log T)^{1/3} + 1$. For this case, we can simple feed the Algorithm 1 with discretization parameter $\varepsilon = 1/(m-1)$. Then, according to the proof of Theorem 1.1, the regret of Algorithm 1 can be bounded as $O(T/m + \sqrt{Tm\log T}) = O(\sqrt{Tm\log T})$ as desired. $\qquad\square$

## C.2 Missing Algorithm and Proof of Theorem B.2

The detailed algorithm description when the number of qualities is large is provided in Algorithm 3.

---

**Algorithm 3:** Algorithm for arbitrary size $m$ of product quality space.

---

1 **Input:** Discretization parameter $\varepsilon$ and pooling precision parameter $\widehat\varepsilon$.

2 **Input:** Instance $\mathcal{I}$ with quality space $\Omega$ and prior $\lambda$.

3 Construct instance $\mathcal{I}^\dagger$ as follows: Let the quality space $\Omega^\dagger = \{\bar\omega_i^\dagger\}_{i \in [\lceil 1/\varepsilon \rceil + 1]}$ where $\bar\omega_1^\dagger = 0, \lambda_1^\dagger = \lambda_1$; and $\bar\omega_{i+1}^\dagger = \mathbb{E}_{\omega \sim \lambda}[\omega \mid \omega \in ((i-1)\widehat\varepsilon, i\widehat\varepsilon]]$, and let the prior $\lambda^\dagger = (\lambda_i^\dagger)_{i \in [\lceil 1/\varepsilon \rceil + 1]}$ where $\lambda_{i+1}^\dagger = \mathbb{P}_{\omega \sim \lambda}[\omega \in ((i-1)\widehat\varepsilon, i\widehat\varepsilon]]$ for all $1 \leq i \leq \lceil 1/\widehat\varepsilon \rceil$.

4 Run Algorithm 1 on instance $\mathcal{I}^\dagger$ with discretization parameter $\varepsilon$.

---

In below, we provide a regret bound that is independent of the size of quality space and it holds for valuation function beyond the additive one as long as it satisfies the following assumption:

**Assumption 2.** *Function $\kappa(p, \cdot)$ satisfies that for any price $p \in [0, U]$, for any $q_1, q_2$ where $q_1 \leq q_2$, $\kappa(p, q_1) - \kappa(p, q_2) \leq q_2 - q_1$.* [7]

Notice that additive valuation $v(\theta, \omega) = \theta + \omega$, which has $\kappa(p, q) = p - q$, satisfies the above assumption.

**Proposition C.1.** *With Assumption 1 and Assumption 2, Algorithm 3 with $\widehat\varepsilon = \varepsilon = (\log T/T)^{1/4}$ has an expected regret of $O(T^{3/4}(\log T)^{1/4})$ independent of the size $m$ of quality space.*

Given the above Proposition C.1, Theorem B.2 simply follows as additive valuation function satisfies Assumption 2.

*Proof of Proposition C.1.* We fix a small $\widehat\varepsilon \in (0, 1)$. Let $\mathcal{I}$ be an instance with quality space $\Omega$ and prior $\lambda \in \Delta^\Omega$. For exposition simplicity, let us assume that for each $i \in [\lceil 1/\widehat\varepsilon \rceil]$, there exists at least one quality $\omega \in \Omega$ such that $\omega \in ((i-1)\widehat\varepsilon, i\widehat\varepsilon]$. We now construct a new instance $\mathcal{I}^\dagger$ with quality space $\Omega^\dagger = (\bar\omega_i^\dagger)_{i \in [\lceil 1/\widehat\varepsilon \rceil + 1]}$ and prior $\lambda^\dagger = (\lambda_i^\dagger)_{i \in [\lceil 1/\widehat\varepsilon \rceil + 1]}$ as follows:

- for $i = 1$: $\bar\omega_i^\dagger = 0, \lambda_i^\dagger = \lambda_1$;

---

[7]We can also relax the assumption to be $\kappa(p, q_1) - \kappa(p, q_1) \leq L(q_2 - q_1)$ where an arbitrary constant $L \in \mathbb{R}^+$ can be treated similarly.

- for $2 \leq i \leq \lceil 1/\widehat{\varepsilon} \rceil + 1$: $\bar{\omega}_i^\dagger = \mathbb{E}_{\omega \sim \lambda}[\omega \mid \omega \in ((i-2)\widehat{\varepsilon}, (i-1)\widehat{\varepsilon}]], \lambda_i^\dagger = \mathbb{P}_{\omega \sim \lambda}[\omega \in ((i-2)\widehat{\varepsilon}, (i-1)\widehat{\varepsilon}]]$.

Essentially, the instance $\mathcal{I}^\dagger$ is constructed by pooling all product qualities that are "close enough" with each other (i.e., qualities in a grid $((i-1)\widehat{\varepsilon}, i\widehat{\varepsilon}]$). By construction, we know that $|\Omega^\dagger| = O(1/\widehat{\varepsilon})$. Given a price $p$ and an advertising $\rho$, let $\mathsf{Rev}_{\mathcal{I}}(p, \rho)$ be the seller's revenue for problem instance $\mathcal{I}$. In below, we have the following revenue guarantee between these two problem instances $\mathcal{I}, \mathcal{I}^\dagger$.

**Lemma C.2.** *Let $p^*, \rho^*$ be the optimal price and optimal advertising for instance $\mathcal{I}$, with Assumption 2, there exists a price $p^\dagger$ and advertising $\rho^\dagger$ for instance $\mathcal{I}^\dagger$ such that $\mathsf{Rev}_{\mathcal{I}}(p^*, \rho^*) \leq \mathsf{Rev}_{\mathcal{I}^\dagger}(p^\dagger, \rho^\dagger) + \widehat{\varepsilon}$.*

The proof of the above Lemma C.2 utilizes Assumption 2 and is provided subsequently. With Lemma C.2, by feeding Algorithm 1 with new instance $\mathcal{I}^\dagger$, the total expected regret for instance $\mathcal{I}$ can be bounded as follows

$$\mathsf{Regret}_{\mathcal{I}}[T] \leq O\left(T\widehat{\varepsilon} + T\varepsilon + \sqrt{|\mathcal{S}|T \log T}\right) = O\left(T\widehat{\varepsilon} + T\varepsilon + \sqrt{\frac{1}{\varepsilon\widehat{\varepsilon}}T \log T}\right) \leq O\left(T^{3/4}(\log T)^{1/4}\right)$$

where the term $T\widehat{\varepsilon}$ is from Lemma C.2 and due to reducing the instance $\mathcal{I}$ to the new instance $\mathcal{I}^\dagger$, the term $T\varepsilon + \sqrt{|\mathcal{S}|T \log T}$ is the incurred regret of Algorithm 1 for the new instance $\mathcal{I}^\dagger$ where the number of discretized types $|\mathcal{S}|$ for the new instance $\mathcal{I}^\dagger$ equals $\frac{1}{\widehat{\varepsilon}\varepsilon}$, and in the last inequality, we choose $\widehat{\varepsilon} = \varepsilon = (\log T/T)^{1/4}$. $\qquad \square$

In below, we provide the proof for Lemma C.2.

*Proof of Lemma C.2.* Let us fix the problem instance $\mathcal{I}$ with quality space $\Omega, |\Omega| = m$ and prior distribution $\lambda$. Let $\mathcal{I}^\dagger$ be the constructed instance (see Line 3 in Algorithm 3). In the proof, we construct a price $p^\dagger$ and an advertising strategy $\rho^\dagger$ for instance $\mathcal{I}^\dagger$ based on $p^*, \rho^*$. Consider a price $p^\dagger = p^* - \widehat{\varepsilon}$. In below, we show that how to construct advertising strategy $\rho^\dagger$ from the advertising strategy $\rho^*$. In particular, for each posterior mean $q \in \mathsf{supp}(\rho^*)$, we construct a corresponding posterior mean $q^\dagger \in \mathsf{supp}(\rho^\dagger)$, and furthermore, with Assumption 1 and Assumption 2, we also show that we always have $\kappa(p^*, q) \geq \kappa(p^\dagger, q^\dagger)$. Recall that from Lemma A.2, the advertising strategy $\rho^*$ satisfies $\{i \in [m] : \rho_i^*(q) > 0\} \leq 2$ for all $q \in \mathsf{supp}(\rho^*)$. Our construction based on threes cases of $\{i \in [m] : \rho_i^*(q) > 0\}$.

- **Case 1** – if $\{i \in [m] : \rho_i^*(q) > 0\} = \{i'\}$, in this case, suppose $\bar{\omega}_{i'} \in ((j-1)\widehat{\varepsilon}, j\widehat{\varepsilon}]$ for some $j \in [\lceil 1/\widehat{\varepsilon} \rceil]$, then consider

$$\rho_{j+1}^\dagger(q^\dagger) = \frac{\lambda_{i'}\rho_{i'}^*(q)}{\lambda_{j+1}^\dagger}; \quad \text{where } q^\dagger = \bar{\omega}_{j+1}^\dagger.$$

From the above construction, we know that $\kappa(p^*, q) = \kappa(p^*, \bar{\omega}_{i'})$, and

$$\kappa(p^*, \bar{\omega}_{i'}) \overset{(a)}{\geq} \kappa(p^\dagger, \bar{\omega}_{i'}) + \widehat{\varepsilon} \overset{(b)}{\geq} \kappa(p^\dagger, \bar{\omega}_{j+1}^\dagger) = \kappa(p^\dagger, q^\dagger)$$

where inequality (a) holds since $\widehat{\varepsilon} = p^* - p^\dagger \leq \kappa(p^*, \bar{\omega}_{i'}) - \kappa(p^\dagger, \bar{\omega}_{i'})$ due to Assumption 1b, and inequality (b) holds since $|\kappa(p^\dagger, \bar{\omega}_{j+1}^\dagger) - \kappa(p^\dagger, \bar{\omega}_{i'})| \leq |\bar{\omega}_{j+1}^\dagger - \bar{\omega}_{i'}| \leq \widehat{\varepsilon}$ due to Assumption 2.

- **Case 2** – if $\{i \in [m] : \rho_i^*(q) > 0\} = \{i', i''\}$ where $i' < i''$, in this case, suppose both $\bar{\omega}_{i'}, \bar{\omega}_{i''} \in ((j-1)\widehat{\varepsilon}, j\widehat{\varepsilon}]$ for some $j \in [\lceil 1/\widehat{\varepsilon} \rceil]$, then consider

$$\rho_{j+1}^\dagger(q^\dagger) = \frac{\lambda_{i'}\rho_{i'}^*(q) + \lambda_{i''}\rho_{i''}^*(q)}{\lambda_{j+1}^\dagger}; \quad \text{where } q^\dagger = \bar{\omega}_{j+1}^\dagger.$$

From the above construction, we know that

$$\kappa(p^*, q) \overset{(a)}{\geq} \kappa(p^\dagger, q) + \widehat{\varepsilon} \overset{(b)}{\geq} \kappa(p^\dagger, \bar{\omega}_{j+1}^\dagger) = \kappa(p^\dagger, q^\dagger)$$

where inequality (a) holds since $\widehat{\varepsilon} = p^* - p^\dagger \leq \kappa(p^*, q) - \kappa(p^\dagger, q)$ due to Assumption **1b**, and inequality (b) holds since $|\kappa(p^\dagger, \bar{\omega}^\dagger_{j+1}) - \kappa(p^\dagger, q)| \leq |\bar{\omega}^\dagger_{j+1} - q| \leq \widehat{\varepsilon}$ due to Assumption 2 and the fact that $q = \frac{\lambda_{i'} \rho^*_{i'}(q)\bar{\omega}_{i'} + \lambda_{i''} \rho^*_{i''}(q)\bar{\omega}_{i''}}{\lambda_{i'} \rho^*_{i'}(q) + \lambda_{i''} \rho^*_{i''}(q)} \in ((j-1)\widehat{\varepsilon}, j\widehat{\varepsilon}]$.

- **Case 3** – if $\{i \in [m] : \rho^*_i(q) > 0\} = \{i', i''\}$ where $i' < i''$, in this case, suppose $\bar{\omega}_{i'} \in ((j'-1)\widehat{\varepsilon}, j'\widehat{\varepsilon}]$ and $\bar{\omega}_{i''} \in ((j''-1)\widehat{\varepsilon}, j''\widehat{\varepsilon}]$ for some $j', j'' \in [\lceil 1/\widehat{\varepsilon}\rceil]$ where $j' < j''$, then consider

$$\rho^\dagger_{j'+1}(q^\dagger) = \frac{\lambda_{i'}\rho^*_{i'}(q)}{\lambda^\dagger_{j'+1}}, \; \rho^\dagger_{j''+1}(q^\dagger) = \frac{\lambda_{i''}\rho^*_{i''}(q)}{\lambda^\dagger_{j''+1}};$$

$$\text{where } q^\dagger = \frac{\lambda^\dagger_{j'+1}\rho^\dagger_{j'+1}(q^\dagger)\bar{\omega}^\dagger_{j'+1} + \lambda^\dagger_{j''+1}\rho^\dagger_{j''+1}(q^\dagger)\bar{\omega}^\dagger_{j''+1}}{\lambda^\dagger_{j'+1}\rho^\dagger_{j'+1}(q^\dagger) + \lambda^\dagger_{j''+1}\rho^\dagger_{j''+1}(q^\dagger)}$$

From the above construction, we know that

$$\kappa(p^*, q) \overset{(a)}{\geq} \kappa(p^\dagger, q) + \widehat{\varepsilon} \overset{(b)}{\geq} \kappa(p^\dagger, q^\dagger)$$

where inequality (a) holds since $\widehat{\varepsilon} = p^* - p^\dagger \leq \kappa(p^*, q) - \kappa(p^\dagger, q)$ due to Assumption **1b**, and inequality (b) holds due to Assumption 2 and the following fact:

$$|q - q^\dagger| = \left| \frac{\lambda_{i'}\rho^*_{i'}(q)\bar{\omega}_{i'} + \lambda_{i''}\rho^*_{i''}(q)\bar{\omega}_{i''}}{\lambda_{i'}\rho^*_{i'}(q) + \lambda_{i''}\rho^*_{i''}(q)} - \frac{\lambda^\dagger_{j'+1}\rho^\dagger_{j'+1}(q^\dagger)\bar{\omega}^\dagger_{j'+1} + \lambda^\dagger_{j''+1}\rho^\dagger_{j''+1}(q^\dagger)\bar{\omega}^\dagger_{j''+1}}{\lambda^\dagger_{j'+1}\rho^\dagger_{j'+1}(q^\dagger) + \lambda^\dagger_{j''+1}\rho^\dagger_{j''+1}(q^\dagger)} \right|$$

$$= \left| \frac{\lambda_{i'}\rho^*_{i'}(q)\bar{\omega}_{i'} + \lambda_{i''}\rho^*_{i''}(q)\bar{\omega}_{i''}}{\lambda_{i'}\rho^*_{i'}(q) + \lambda_{i''}\rho^*_{i''}(q)} - \frac{\lambda_{i'}\rho^*_{i'}(q)\bar{\omega}^\dagger_{j'+1} + \lambda_{i''}\rho^*_{i''}(q)\bar{\omega}^\dagger_{j''+1}}{\lambda_{i'}\rho^*_{i'}(q) + \lambda_{i''}\rho^*_{i''}(q)} \right|$$

$$\leq \frac{\lambda_{i'}\rho^*_{i'}(q)|\bar{\omega}^\dagger_{j'+1} - \bar{\omega}_{i'}| + \lambda_{i''}\rho^*_{i''}(q)|\bar{\omega}^\dagger_{j''+1} - \bar{\omega}_{i''}|}{\lambda_{i'}\rho^*_{i'}(q) + \lambda_{i''}\rho^*_{i''}(q)}$$

$$\leq \frac{\lambda_{i'}\rho^*_{i'}(q)\widehat{\varepsilon} + \lambda_{i''}\rho^*_{i''}(q)\widehat{\varepsilon}}{\lambda_{i'}\rho^*_{i'}(q) + \lambda_{i''}\rho^*_{i''}(q)} = \widehat{\varepsilon}$$

We also note that by construction, for any posterior mean $q \in \mathsf{supp}(\rho^*)$, the corresponding constructed posterior mean $q^\dagger \in \mathsf{supp}(\rho^\dagger)$ satisfies that

$$\rho^\dagger(q^\dagger) = \sum_{i \in [\lceil 1/\widehat{\varepsilon}\rceil + 1]} \rho^\dagger_i(q^\dagger)\lambda^\dagger_i = \rho^*(q) \tag{10}$$

Armed with the above observation $\kappa(p^*, q) \geq \kappa(p^\dagger, q^\dagger)$, we are now ready to show $\mathsf{Rev}_{\mathcal{I}}(p^*, \rho^*) \leq \mathsf{Rev}_{\mathcal{I}^\dagger}(p^\dagger, \rho^\dagger) + \widehat{\varepsilon}$:

$$\mathsf{Rev}_{\mathcal{I}}(p^*, \rho^*) - \mathsf{Rev}_{\mathcal{I}^\dagger}(p^\dagger, \rho^\dagger) = p^* \int_q \rho^*(q)D(\kappa(p^*, q))dq - p^\dagger \int_{q^\dagger} \rho^\dagger(q^\dagger)D(\kappa(p^\dagger, q^\dagger))dq^\dagger$$

$$\overset{(a)}{\leq} p^* \int_q \rho^*(q)D(\kappa(p^*, q))dq - p^* \int_{q^\dagger} \rho^\dagger(q^\dagger)D(\kappa(p^\dagger, q^\dagger))dq^\dagger + \widehat{\varepsilon}$$

$$= p^* \left( \int_q \rho^*(q)D(\kappa(p^*, q))dq - \int_{q^\dagger} \rho^\dagger(q^\dagger)D(\kappa(p^\dagger, q^\dagger))dq^\dagger \right) + \widehat{\varepsilon}$$

$$\overset{(b)}{\leq} \widehat{\varepsilon}$$

where inequality (a) holds since we have $p^\dagger = p^* - \widehat{\varepsilon}$, and inequality (b) holds by the observation $\kappa(p^*, q) \geq \kappa(p^\dagger, q^\dagger)$ and (10). $\qquad \square$