# OpenReview forum: "Dynamic Pricing and Learning with Bayesian Persuasion"
_NeurIPS.cc/2023/Conference — NeurIPS 2023 poster_

### Official Review · Reviewer_p9NT · 2023-07-06

**Soundness:** 3 good
**Presentation:** 2 fair
**Contribution:** 2 fair
**Rating:** 6
**Confidence:** 3

**Summary:**

The paper studies a dynamic pricing model augmented by advertisement campaigns. The problem is framed in a Bayesian persuasion framework, where the authors focus on the online learning version of the model in which the agent is not aware of buyer the demand function from which its types are drawn.

**Strengths:**

The paper studies an interesting model of dynamic pricing, which leans on the Bayesian persuasion framework. Due to the continuity/size of the action space, standard techniques cannot be applied and the results presented seems technically interesting.
Moreover the paper is mostly clear and easy to follow and does a good job at motivating the model.

**Weaknesses:**

I found the description of the discretisation (line 308) a bit hard to follow. A little bit more intuitive discussion on this would be appreciated  for this part.
See the question on the lower bound in the Question section of the review
If convinced on this point I’m willing to revise my score

**Questions:**

1) would revelation like principle hold in this context? Could the space of signals be made binary as the action set or not? A brief discussion on this would be interesting for the BP readers.
2) The types in this problem are stochastic. What if they would be drawn by an adversary? Most of the online BP literature studies the adversarial problem.
3) the authors cite a T^{2/3} lower bound for the dynamic pricing problem. I guess that the your extended model reduces to the standard DP model when the utilities are zero sum and thus the sender commits to uniformly random signalling schemes and thus the posterior would be equal to the prior. Am I correct? If so, I would like to be convinced that zero sum utilities between seller and buyer can be achieved in your model. Moreover more recent UB for the online BP problem with adversarial types have been found, and guarantee a sqrt(T) regret. The work I’m referencing is [Bernasconi, Martino, et al. "Optimal Rates and Efficient Algorithms for Online Bayesian Persuasion." (2023).] which I think should be discussed.

**Limitations:**

Yes

---

> ### Author Rebuttal · Authors · 2023-08-08
>
> Thank you for your comments, we will incorporate your comments and presentation suggestions in the revision. We will respond the questions as follows:
>
> **Q1 (Revelation principle on binary signals):**
> If buyer’s type were fixed, then binary-signal would indeed suffice, as known from previous literature. However, in our model where a buyer's private type is randomly drawn from a distribution, binary-signal does not suffice to achieve optimal advertising in this Bayesian setting. We can show that the gap between the revenue using the binary-signal and the optimal revenue can be large.
>
> As shown in KG-AER’11, when the receiver (i.e, buyer) has private information (i.e., type), the revelation principle claims that there exists a mechanism that the receiver will find it optimal to truthfully reveal their private type to the sender, and then the sender will issue a binary signal to the receiver (given receiver has binary action). Notably, under this mechanism, the buyer will firstly report her private type, which is not the model considered in this work.
>
> Thank you for the suggestion, and we will provide further clarity on this point in our revision.
>
>
> **Q2 (Adversary buyer valuations):**
> In this work, we focus on stochastic setting. Though this is beyond the scope of current paper, we agree with the reviewer that adversary valuation is certainly an interesting direction to explore in future work.
>
> **Q3 (Lower bound and whether zero sum utilities between seller and buyer):**
> The lower bound $\Omega(T^{2/3})$ holds since our setting can recover the standard dynamic pricing problem (Kleinberg and Leighton, FOCS’03) when the prior distribution of the product quality is a point mass, i.e., the product quality is known to both players. In that case, the seller only needs to decide the price, and any signaling scheme does not affect the outcome. Therefore, the seller’s problem is a pure DP with a regret lower bound $\Omega(T^{2/3})$.
>
> We are not sure if we understand what reviewer is referring to as “zero sum utilities”. As in dynamic pricing, in our setting we are optimizing only the seller’s objective which is the revenue function. Buyer is responding by maximizing the valuation function, which depends only on private type (as in dynamic pricing) when the product quality is fixed (i.e. when prior distribution is a point mass). Thus, the connection of our model to the zero-sum game is unclear. Please let us know if the above response do not answer the question, and we will be happy to offer more clarifications.
>
> Thank you for pointing to that reference, we will discuss its relevance in the revision.

---

> > ### Comment · Reviewer_p9NT · 2023-08-12
> >
> > Thank you for responding to my comments. I now understand better the $T^{2/3}$ lower bound. Even if would seem unnecessary I would suggest to include a formal statement (or a brief discussion that extends the construction you presented in the response) in the appendix. My comment on zsg was trying a different reduction that can now be ignored. I'll raise my score accordingly

---

### Official Review · Reviewer_khHp · 2023-07-07

**Soundness:** 4 excellent
**Presentation:** 3 good
**Contribution:** 3 good
**Rating:** 7
**Confidence:** 3

**Summary:**

This work focuses on advertising and selling markets under the Bayesian persuasion framework. On the one hand, this paper models the effect of the advertisement strategy on customers’ beliefs and purchase decisions. On the other hand, the authors also address the problem of online learning the optimal advertisement strategy via learning the buyers’ type distribution and show $O(T^{2/3} m^{1/3})$ regret for discrete quality space and $O(T^{3/4})$ regret for arbitrary quality space.

**Strengths:**

The problem and model discussed in this paper are intriguing, and the authors devise a novel quality-and-price-dependent discretization scheme for proving the learning result. Notably, the learning process only necessitates observations of the buyers' purchase decisions, eliminating the need for explicit knowledge of their individual types. This work may be of interest to the community.

**Weaknesses:**

Is the value function required to be linear in $\omega$? I notice that using $q$ for simplification relies on this linearity assumption. Additionally, in order to determine the critical type, the seller would typically need prior knowledge of the buyers' value function, which may not always be feasible. Is it possible to relax this requirement within this framework?

**Questions:**

1. Are the regret bounds of $O(T^{2/3})$ and $O(T^{3/4})$ tight for discrete and continuous quality spaces, respectively?
2. Is there a deeper understanding of the optimization program P_opt? Can it provide insights into the phenomenon where revealing more or less information about the product can sometimes lead to increased revenue?

**Limitations:**

No concerns here.

---

> ### Author Rebuttal · Authors · 2023-08-08
>
> Thank you for your comments, we will incorporate your comments in the revision. We start with addressing the technique comment:
>
> **Q1 (Tightness of $O(T^{2/3})$ and $O(T^{3/4})$ for discrete and continuous quality spaces):**
> Our regret $O(T^{2/3})$ for discrete quality space is tight, please see the discussions from Line 129-132. However, we do not have a matching lower bound for continuous quality space, and we leave it as an interesting future work direction.
>
> **Weakness (assumptions):**
> In current work, we assume the buyer’s value is linear w.r.t the quality. This assumption is widely-adopted in current literature and is also natural in many economic applications (see Line 114 for references). We believe that the assumption on requiring knowing valuation function can be relaxed by further using certain bandit learning techniques. Fully relaxing these assumptions is an interesting technical direction for future work.
>
>
> **Q2 (Understanding program P_opt):**
> We have some examples showing that partial information revealing can indeed increase the seller’s revenue. In particular, we can derive closed-form characterization for the optimal pricing and advertising strategy, and use it to show some qualitatively results under some canonical instances (e.g., binary quality, log-concave type distributions). However, we currently do not have closed-form characterization regarding program $P_\text{opt}$ for general instances. We will add discussions in the revision.

---

> > ### Comment · Reviewer_khHp · 2023-08-18
> >
> > Thanks to the authors for addressing my concerns. I have no further questions.

---

### Official Review · Reviewer_mNn4 · 2023-07-08

**Soundness:** 3 good
**Presentation:** 3 good
**Contribution:** 3 good
**Rating:** 6
**Confidence:** 3

**Summary:**

The authors consider an online single product dynamic pricing product, where a buyer's purchase decision is not only influenced by the offered price from the selling, but also depends on the advertising strategy used by the seller. The authors formalized the latter via the model of Bayesian persuasion framework, which is a well establish model in the economic literature (to the best of my knowledge). In the only problem, the seller is uncertain about the type distribution of a customer, but the seller knows the probability distribution on the quality of a product (which is a random variable). The authors' main results is the design and analysis of a UCB algorithm, where the authors crucially use a mild assumption on the customer's utility to reduce the size of decision variables on the advertising strategy. The regret  bound derived is nearly tight (module a factor of $m^{1/3}$, where $m$ is the number of possible quality. )

**Strengths:**

- The setting and the algorithm design, analysis seem novel, and this is the first piece of work on online revenue mgmt with Bayesian persuasion to my knowledge.

- Despite the complexity of the set up and the algorithm, the authors provide enough intuitions for readers to understand the content, and I feel that the proposed work could inspire further interest in other pricing problems.

- The whole algorithm framework is well crafted such that the tightness on $T$ is achieved.

**Weaknesses:**

- If I understand the model set up correctly (please also see Question (1) in "Questions"), the seller seems to need to know $\mu_t$ that relates to the time t customer's perception on a product's quality given the seller's advertising strategy. This assumption could appear a bit strong, since in most online pricing setting the learning agent should be uncertain about a customer's purchase characteristics.

- I find the modeling assumption on the product quality being random a little strange. Putting myself as a seller who can observe the quality of a product before an action, it could seem natural for myself to sell products of high quality first (let say if I see a low quality product, then I will just forgo it, and repeat until I can find a high quality product for the customer) in order to yield a higher revenue? If the seller could not observe the quality of a product, I do find modeling the unseen quality as a random variable reasonable.

- There is a looseness in $m$, but I regard it as a minor weakness given the complexity of the problem.

- There is a lack of numerical results, which could yield some immediate intuitions.

**Questions:**

(1) Can the authors explicit confirm if the seller possesses the knowledge on $\mu_t$, the posterior distribution of time $t$ customer's perceived quality conditional on an advertisement strategy? Just to double confirm my understanding, in the reduction from selecting $\sigma_t$ to the selection of $\rho_t$ which is a uni-variate probability distribution, the reduction involves considering the mean of a mixture of different $\mu_t$ distribution. Thus, in order to arrive with the desired $\rho_t$, we need to have the correct mixture, which involves knowing $\mu_t$. Can I please check if my understanding above is correct?

(2) Can the author improve the dependence on $T$ if we have a discrete set of (say $K$) prices?

(3) If my understanding in Question (1) is correct, could there be a way to extend the framework to the case when $\mu_t$ is stationary across time but not known to the seller, or is there a $\Omega(T)$ regret lower bound due to the insufficiency in the information in the outcome?



**Limitations:**

There is no negative societal impact, this is a theoretical work.

---

> ### Author Rebuttal · Authors · 2023-08-08
>
> Thank you for your comments, we will incorporate your comments in the revision. We below respond the questions as follows:
>
> **Q1 and Q3 (The seller’s knowledge of buyer’s posterior belief $\mu_t$):**
> The reviewer is correct that the seller knows the (realized) buyer's posterior belief $\mu_t$. We would like to clarify that this knowledge isn't assumed but derived. Specifically, in our setting, seller designs the signaling scheme and can observe the realized signal $\sigma_t$, and the incoming buyer shares the same prior distribution with the seller (i.e., the prior does not depend on buyer’s private type). Thus, the seller can infer the buyer’s Bayesian posterior belief $\mu_t$ upon realizing a signal $\sigma_t$. We appreciate the reviewer's insight and will make it more clear in our revision.
>
>
> **Weakness 2 (Modeling assumption on random quality):**
> We are not sure if we completely understand reviewer’s comment. We would like to clarify that it is a priori unknown whether a product quality is particularly low or high as the buyer’s valuation also depends on buyer’s private type, which is unknown to the seller, e.g., a seemingly ”low“ quality product could still lead to a high valuation if buyer’s type is sufficiently high, and thus the seller can still sell this ”low“ quality product. Please let us know if the above response do not answer the question, and we will be happy to offer more clarifications.
>
>
> **Q2 (Potential improvement for $K$ prices):**
> If the seller only has $K$ prices, we think that the regret bound cannot be immediately improved based on our current analysis. The challenge is as follows: even though the seller has only $K$ prices, the seller’s problem is still a continuous-armed bandit problem due to the rich space of the possible advertising strategies (thus a carefully designed discretization over type space is still needed), we will add discussions in the revision.

---

### Official Review · Reviewer_78HV · 2023-07-14

**Soundness:** 3 good
**Presentation:** 2 fair
**Contribution:** 3 good
**Rating:** 5
**Confidence:** 4

**Summary:**

The paper extends the traditional dynamic pricing model (e.g. Kleinberg and Leighton) to also include product quality. In each period, we get both a random type of the buyer as well as a random quality of the product. The seller can employ an advertising strategy, which is a mapping from product quality to a signal, which must be decided in each step before the signal is realized. To be more precise, in each period:
* buyer type is drawn $\theta_t \sim F$
* seller decides on a signaling scheme mapping item quality $\omega$ to a signal $\sigma$
* seller also decides on a price $p_t$ for the product
* item quality $\omega \sim \lambda$ is realized. Signal $\sigma$ is computed and sent to the buyer:
* buyer decides to purchase if $\mathbb{E}[\theta, \omega) \mid \sigma] \geq p_t$


The seller knows all parameters of the model except $F$;. The goal of the seller is to learn $F$ and compete against the algorithm that knows $F$ and chooses the prices and advertising strategies optimally in each period. Naive MAB approaches are too high dimensional, so the authors use the following clever idea: define the threshold type $\kappa(p_t, \rho_t)$ which is the threshold specifying above which types they buy. Now, If we choose the signaling scheme carefully we can observe whether $\theta$ was above or below $\kappa(p_t, \rho_t)$. with that, we can build an estimate of $F$. The question now becomes discretizing the prices and advertising strategies appropriately. Once the estimate is built, we use the UCB estimates to solve a program and allocate according to that.

The algorithm has $O(T^{2/3})$ regret due to two effects: (i) price discretization; (ii) explore and exploit.

**Strengths:**

* Interesting model mixing persuasion and pricing. This is a nice and natural extension of the pricing model.

* There is a clever idea: instead of discretizing in the space of signals, we discretize prices and estimate $F$ directly.

* The results match the lower bound of Kleinberg-Leighton. On that, the authors should probably discuss more why the Kleinberg-Leighton lower bound applies to this problem. I imagine their setting corresponds to known quality, so we don’t need to signal anything. I recommend saying this explicitly.


**Weaknesses:**

* In terms of motivation, it is unclear that we change the advertising so fast. Typically it takes a few iterations with the signaling scheme for a buyer to learn the mapping from signals to actions. Now that the buyer is seeing a new one in each period, I don't see how this would translate to a practical situation.


* Mention "linear" in Assumption 1a to make it clear. From reading assumption 1, I was under the impression that we were just assuming $v$ to be monotone on $\omega$. Btw: I think you mean "affine" instead of linear. The function $v(\theta, \omega) = \theta + \omega$ is not linear (in the usual definition of linear)

* I think Explore-then-Exploit are underwhelming in terms of ML algorithms. You can always learn the parameters in an explore phase and just optimize. There are non-trivial many aspects of the error analysis, but in terms of algorithmic techniques, I found it to be limited.

* The paper gets too long to get to the actual algorithm with many discussions that could be done much faster. For example, teh entire discussion in Section 2.1 is the Splitting Lemma of Aumann-Maschler.

**Questions:**

See my comments in the last box.

**Limitations:**

No concerns.

---

> ### Author Rebuttal · Authors · 2023-08-08
>
> Thank you for your comments, we will incorporate your comments and presentation suggestions in the revision. We below mainly address the technique comment:
>
> **Weakness 3 (Why not Explore-then-Exploit algorithm):**
>
> We believe the regret achievable by any "Explore-then-Exploit" algorithm is $\tilde{O}(T^{3/4}m^{1/4})$, which is suboptimal compared to our algorithm’s regret $\tilde{O}(T^{2/3}m^{1/3})$. The suboptimality of "Explore-then-Exploit" can be demonstrated as follows:
> 1. **In exploration phase**: By fixing a pre-specified discretized type space $\mathcal{S}$ and choosing different prices with no-information advertising, we can obtain estimates for the demand function on points in $\mathcal{S}$. If we can have a reasonably good estimate for the demand function $D$ on the points in $\mathcal{S}$, then we can construct an approximately-optimal pricing and advertising strategy, and use it in exploitation phase. The per-round regret in this phase is constant, with a total of $|\mathcal{S}|\cdot \frac{\log T}{\varepsilon^2}$ rounds (by Hoeffding inequality) to obtain $\varepsilon$-accurate estimated demand, i.e., $|D^{\text{estimated}}(x) - D(x)|\le\varepsilon, \forall x\in\mathcal{S}$;
> 2. **In exploitation phase**: To get an $\varepsilon$-approximate optimal pricing and advertising strategy using the estimated demand, we can construct the set $\mathcal{S}$ as proposed in the paper, yielding $|\mathcal{S}|=\frac{m}{\varepsilon}$.
>
> Thus, the total regrets is: $\frac{m}{\varepsilon}\cdot \frac{\log T}{\varepsilon^2} + \varepsilon \cdot \left(T - \frac{m}{\varepsilon}\cdot\frac{\log T}{\varepsilon^2}\right)=\tilde{O}(T^{3/4}m^{1/4})$ with optimizing $\varepsilon = (m/T)^{1/4}$.

---

### Official Review · Reviewer_kmXq · 2023-07-18

**Soundness:** 3 good
**Presentation:** 2 fair
**Contribution:** 3 good
**Rating:** 6
**Confidence:** 4

**Summary:**

In this work, the authors study a novel setup of repeated posted-price auctions, where a single seller deals with a single myopic buyer (or, equiv, iid new buyer in each round). In each round, a new good is offered and has characteristics that affect buyer valuation (so, goods are not equal between rounds). The novelty of the setup consists in introducing a new interaction step, where the seller reveals characteristics of each good through some signals that might improve perception of the buyer. The buyer is aware of this step, which is referred to as advertising (due to analogy with cases in practice).
So, in each round the seller submits two inputs (price, advertising signal), while the buyer reacts by a decision: accept or reject the price for this good.

The authors clearly argue why this setup differs from earlier considered ones (either price-only, or advertising-only). They provide a novel algorithm (which is announced in advance. before the game) and prove that this algorithm has regret upper bound which is slightly above the lower bound by the factor of log T (for the setup without advertising).

**Strengths:**

- novel setup in the area of repeated auctions with plenty opportunity for future work
- new algorithm with theoretical guarantees
- easy understandable basic ideas behind the proofs

**Weaknesses:**

- presentation (a lot of space for improvement)
- some questions to setup and setting
- lack of analysis and discussion of proposed algorithm (computational complexity, limitations in practice, ...)
(see “Questions” for details)

**Questions:**

A. Setting questions and other matter stuff:

1. Line 70 about definition of Omega: Any extentions to multidimentional \Omega?

2. Footnote 3: “ This is motivated by the fact that at each round, the buyer is facing a fresh product, whose quality is drawn independently across time.” I would add that this is not just because of it. But, also because, in each round, we consider a fresh buyer (myopic to the other rounds). This setting might be changed even in the case of different products. For instance, see settings with a single buyer that makes strategic decisions to optimize his/her utility over all rounds in the cases of different products: [Zhiyanov2020, Drutsa2020] (by the way, these works are extensions of the work [39] Kleinberg et al 2003 you are citing).
So, I would suggest upgrading your footnote by adding these thoughts and probably indicate some future work about injecting the strategic buyer in your setting.

3. Lines 96-97 “and a static price and advertising strategy maximizes total expected revenue over the time horizon T” Hard to parse this sentence. It sounds like, static (p, \phi) are the best solutions over all possible ({p_t} , {\phi_t}). The authors seem to state that these are the best among static p_t, \phi_t. If yes, please reformulate the sentence to be clearer

4. Formula after Line 99:
Over what is expectation taken right after the \Sum sign? It looks like all randomness is under the hood of Rev()..
Or, ... do we expect mixed (non-pure) strategies of the Seller?

5. Line 102 “of the (discrete) product quality space.” What will be when \Omega is not discrete? E.g., continuous?

6. Footnote 4 “ arbitrary Lipschitz constant can be treated similarly.” It would be nice to understand how this constant participates in the upper bound .. E.g. in workds [Mao2018, Drutsa2020], it plays some notable role and shows severe dependance on the degree of Lipschitz assumption. Is it true in your case?

7. Algorithm 1. Line 5. “ computed as an optimal solution to program P^{UCB}.” What are the costs for finding optimal solution in P_t^{UCB}? Overall, it would be nice to clearly address complexity of the algorithm in a separate paragraph in the text with discussions.


B. Presentation/wording issues:

1. Lines 14-17. It is written “the valuation function is linear in the product quality” and “This result requires some natural monotonicity and Lipschitz assumptions on the valuation function…”. It is unclear: linear valuation is Lipschitz by definition... Or result is about non-parametric valuation.. ?
Later, in the main text, I found that valuation depends on 2 parameters, where properties on linearity and Lipschitz are on different parameters. I suggest rewriting Line 17 to make it clearer from abstract.

2. Structure of the paper. I suggest to improve structure of the work. At least
- By splitting Inroduction in separate Sections: Intro, Problem formulation (statement)
- By adding conclusion

3. Line 70 “with a public prior distribution \lambda on.. ” Hard to parse. Only in Line 78, I've got that lambda is ex-ante for the whole game play (before round 1).

4. Lines 81-82: “if the expected valuation” Why is this the reasonable way to decide? I mean this way to transform quality distribution to a number for comparison with price? Maybe, Median, Moda, Quantiles are better?
Is it because Buyer maximizes some utility? If yes, what is the utility in this case?
Later I found the answer in Lines 215-217. However, I'd suggest improving the text in Page2 by adding the info that the setting is about Bayesian rational buyer

5. Line 287: “we can only use estimates of demand function on a discretized type space, say S” Hard to read. Please, simplify.

6. Algorithm 1. Line 2 “any no”. I do not understand "any no".

7. Lines 353-358 (and many other earlier): Text here repeats earlier stated (several times) assumptions and reference of results to Appendices.
It is better to save space and show more details on proofs, limitations, computational complexity, and conclusions...


C. Minor issues:
- Lines 19-20 “the regret lower bound for dynamic pricing within logarithmic factors, which is a special case of our problem.” Wording: hard to parse what is the special case: factor or pricing. Please, rewrite
- Line 167: “the these” -> “these”?
- No line counts between Lines 217-218
- Line 286: “to solve to find” -> “to find”?



LITERATURE:
[Mao2018] Mao,  Leme, and Schneider “Contextual pricing for lipschitz buyers” NeurIPS 2018.
[Zhiyanov2020] Zhiyanov and Drutsa "Bisection-based pricing for repeated contextual auctions against strategic buyer." ICML  2020.
[Drutsa2020] Drutsa "Optimal non-parametric learning in repeated contextual auctions with strategic buyer." ICML 2020.




========

AFTER REBUTTAL
I thank the authors for answering several questions from my list. I hope that the answers will be reflected and fully addressed in the new revision of the work - including those questions that remain without comments from the authors.

**Limitations:**

No. I would suggest adding a paragraph about computational complexity analysis.

---

> ### Author Rebuttal · Authors · 2023-08-08
>
> Thank you for your comments, we will incorporate your comments and presentation suggestions in the revision, and also adequately discuss the relevance to the pointed literature. We below respond the important questions:
>
> **Weakness 3 and Q7 (Complexity of the algorithm):**
> We kindly remind the reviewer that we indeed mention the complexity of our algorithm (please see Line 340-342 right below the Algorithm 1). In particular, it can be shown that the empirical program $P_t^{\text{UCB}}$ is a convex program where the Bayes-constraints can be reformulated as convex constraints (Arieli, Itai, et al. EC’20, Candogan EC’19), thus an efficient algorithmic solution exists to solve $P_t^{\text{UCB}}$. We make this argument more formally in Proposition A.1 (due to space limit, we put it in the appendix). We will move it to the main text in the revision.
>
> **Q5:** In Line 102, the regret bound $O(T^{2/3}(m\log T)^{1/3})$ holds for discrete state space where $m$ is the number of states. For continuous state space, we have regret $\tilde{O}(T^{3/4})$ (please see Line 125-127 and our Theorem C.2).
>
> **Q6 (Lipschitz constant):** With the Lipschitz constant $L$ of the value function $|v(\theta_1, \omega)-v(\theta_2, \omega)|\le L|\theta_1-\theta_2|$, our regret bound will be $\tilde{O}(T^{2/3}(Lm)^{1/3})$ for discrete state space, and be $\tilde{O}(L^{1/4}T^{3/4})$ for continuous state space. We will clarify it in the revision.

---

### Decision · Program_Chairs · 2023-09-21

**Decision:**

Accept (poster)

**Comment:**

The reviewing team was in agreement positive about this paper. All reviewers liked the contribution of the paper, and made some comments about improving the presentation which I believe can be implemented for a final version. I also liked that the paper's upper bound for the discrete case is *tight* to O(T^{2/3}), despite the fact that this can be achieved by an explore-then-commit algorithm.